# Dilation of fusion pores by crowding of SNARE proteins

Zhenyong Wu[1,2], Oscar D Bello[2,3†], Sathish Thiyagarajan[4†], Sarah Marie Auclair[2,3], Wensi Vennekate[1,2], Shyam S Krishnakumar[2,3], Ben O'Shaughnessy[5], Erdem Karatekin[1,2,6,7]*

[1]Department of Cellular and Molecular Physiology, School of Medicine, Yale University, New Haven, United States; [2]Nanobiology Institute, Yale University, West Haven, United States; [3]Department of Cell Biology, School of Medicine, Yale University, New Haven, United States; [4]Department of Physics, Columbia University, New York, United States; [5]Department of Chemical Engineering, Columbia University, New York, United States; [6]Molecular Biophysics and Biochemistry, Yale University, New Haven, United States; [7]Laboratoire de Neurophotonique, Université Paris Descartes, Faculté des Sciences Fondamentales et Biomédicales, Centre National de la Recherche Scientifique, Paris, France

*For correspondence: erdem. karatekin@yale.edu

†These authors contributed equally to this work

Competing interests: The authors declare that no competing interests exist.

**Abstract** Hormones and neurotransmitters are released through fluctuating exocytotic fusion pores that can flicker open and shut multiple times. Cargo release and vesicle recycling depend on the fate of the pore, which may reseal or dilate irreversibly. Pore nucleation requires zippering between vesicle-associated v-SNAREs and target membrane t-SNAREs, but the mechanisms governing the subsequent pore dilation are not understood. Here, we probed the dilation of single fusion pores using v-SNARE-reconstituted ~23-nm-diameter discoidal nanolipoprotein particles (vNLPs) as fusion partners with cells ectopically expressing cognate, 'flipped' t-SNAREs. Pore nucleation required a minimum of two v-SNAREs per NLP face, and further increases in v-SNARE copy numbers did not affect nucleation rate. By contrast, the probability of pore dilation increased with increasing v-SNARE copies and was far from saturating at 15 v-SNARE copies per face, the NLP capacity. Our experimental and computational results suggest that SNARE availability may be pivotal in determining whether neurotransmitters or hormones are released through a transient ('kiss and run') or an irreversibly dilating pore (full fusion).

## Introduction

Fusion pores are nanoscale connections between membrane-enclosed compartments that are key intermediates during membrane fusion reactions such as the exocytotic release of neurotransmitters and hormones (*Lindau et al., 2003*). Following nucleation by specialized proteins (*Chernomordik and Kozlov, 2008*), fusion pores flicker repeatedly and then dilate or reseal during the release of hormones (*Lindau et al., 2003*) or neurotransmitters (*Alabi and Tsien, 2013*; *He and Wu, 2007*; *Staal et al., 2004*), or during fusion mediated by viral proteins (*Cohen and Melikyan, 2004*). The mechanisms that govern these behaviors are poorly understood, despite the availability of sensitive electrical and electrochemical methods to detect single fusion pores during protein-free fusion (*Chanturiya et al., 1997*; *Mellander et al., 2014*), viral-protein-induced fusion (*Cohen and Melikyan, 2004*), and exocytosis (*Lindau, 2012*). Even the very nature of the fusion pore intermediate (whether lipid- or protein-lined) is debated (*Bao et al., 2016*).

During exocytotic neurotransmitter or hormone release, a fusion pore opens as vesicle-associated soluble N-ethylmaleimide-sensitive factor attachment protein receptors (v-SNAREs) pair with

cognate t-SNAREs on the target plasma membrane (*Südhof and Rothman, 2009*). This is a tightly regulated process that requires the coordinated actions of several proteins, including Munc18, Munc13, and others (*Rizo and Xu, 2015*). Complex formation between the v- and t-SNAREs is likely to start at the membrane distal N-termini and may proceed in stages toward the membrane-proximal regions (*Gao et al., 2012*). Assembly of the SNARE domains results in a four-helix bundle (SNAREpin) that brings the two bilayers into close proximity, but assembly is thought to be halted at some stage to poise vesicles for fast release. Calcium influx in response to depolarization is thought to lead to further SNARE assembly that promotes pore nucleation. This last step — coupling calcium entry to fusion — also requires Synaptotagmin and Complexin, which may actively contribute to pore opening. The initial fusion pore is a metastable structure that may reseal without ever dilating beyond ~1–2 nm in size. This results in transient 'kiss and run' exocytosis, a well-established mode of fusion for hormone-secreting cells (*Alabi and Tsien, 2013*; *Hanna et al., 2009*; *Fulop et al., 2005*). By contrast, whether transient fusion is a relevant mode of release for synaptic vesicle fusion has been a subject of debate (*Alabi and Tsien, 2013*; *He and Wu, 2007*; *Staal et al., 2004*; *Pawlu et al., 2004*; *Chapochnikov et al., 2014*), mainly because of technical challenges in probing fusion pores directly during synaptic release. For both neuronal and endocrine release, little is known about the molecular mechanisms that govern pore dilation and that set the balance between transient and full fusion (*Alabi and Tsien, 2013*), in large part due to a lack of biochemically defined assays that are sensitive to single-pores.

Here, using a recently developed nanodisc-cell fusion system (*Wu et al., 2016*), we found that the presence of just a few SNARE complexes can nucleate a pore, but that reliable pore dilation necessitates many more.

## Results

### Fusion between v-SNARE reconstituted nanolipoprotein particles and flipped t-SNARE cells

We used 21–27 nm diameter nanolipoprotein particles (NLPs) (*Bello et al., 2016*) to determine whether SNAREs alone can catalyze pore dilation. By contrast, most previous studies employed much smaller, 6–18-nm diameter nanodiscs (NDs) (*Bao et al., 2016*; *Wu et al., 2016*; *Shi et al., 2012*), whose dimensions restricted pore diameters to ≤4 nm (*Wu et al., 2016*) and SNARE copy numbers to ≤9, precluding studies of pore dilation (*Bello et al., 2016*). We incorporated v-SNAREs into NLPs that were stabilized by a recombinant apolipoprotein E variant consisting of the N-terminal 22-kDa fragment (ApoE422k). We varied the lipid-to-ApoE422k ratio to control the NLP size and the VAMP2-to-ApoE422k ratio to tune the number of v-SNARE copies per NLP (*Bello et al., 2016*) (*Figure 1*). We confirmed that vNLPs fused with liposomes that were reconstituted with t-SNAREs in a SNARE-dependent manner using a previously described bulk assay that monitors calcium release through pores connecting v-SNARE nanodiscs with t-SNARE liposomes (*Bello et al., 2016*; *Shi et al., 2012*) (*Figure 1—figure supplement 1a*). Although NLP pores could in principle grow to ≥10 nm in diameter (Figure 4b), much larger than the ~4 nm allowed by the membrane scaffold protein (MSP) based small ND geometry (*Wu et al., 2016*; *Shi et al., 2012*), bulk calcium release rates were comparable between vNLP and vMSP NDs loaded with similar v-SNARE copy numbers (*Figure 1—figure supplement 1b*), confirming that the bulk assay is largely insensitive to pore properties under these conditions (*Bello et al., 2016*).

We then confirmed lipid mixing between the membranes of vNLPs and flipped t-SNARE cells (tCells) using a previously described protocol (*Wu et al., 2016*) (*Figure 2*). NLPs co-labeled with one mole % each of DiI (donor) and DiD (acceptor) were incubated with tCells for 30 min at 4°C, a temperature that allows docking but not fusion. Cells were then rinsed to remove free NLPs and mounted onto a confocal microscope stage held at 37°C to initiate fusion and imaging of DiI and (directly excited) DiD fluorescence. At the concentrations used, the DiI fluorescence is quenched by DiD when the dyes are initially in the NLP membrane. Upon fusion, the dyes become diluted in the plasma membrane and the DiI fluorescence increases. Directly excited DiD fluorescence provides a measure of the amount of docked NLPs. The ratio of the DiI to DiD intensity normalizes the lipid-mixing signal to the amount of docked NLPs. Normalized lipid-mixing signals increased when tCells were incubated with vNLPs carrying eight v-SNAREs total (vNLP8), but not with empty NLPs (eNLP)

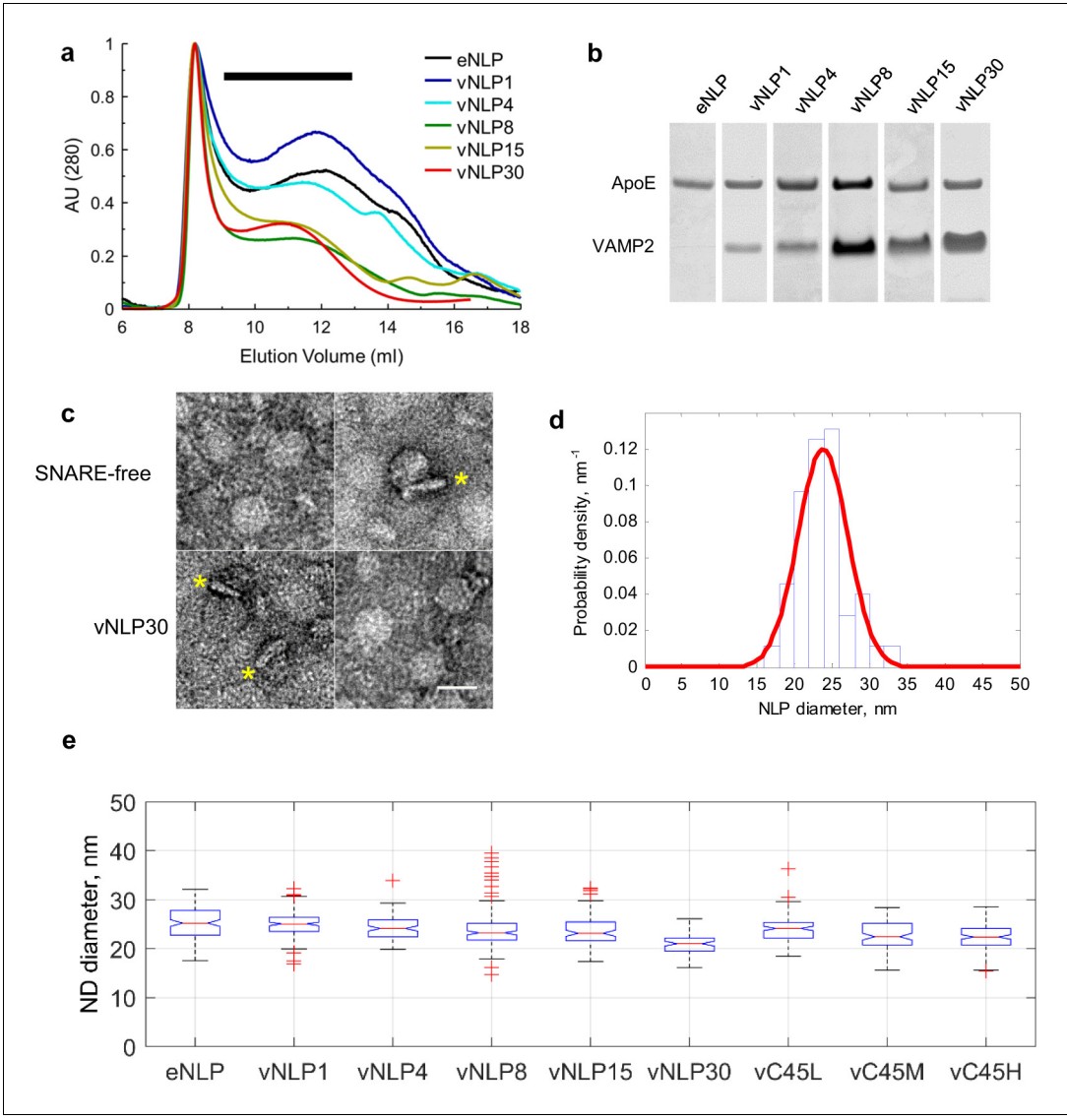

**Figure 1.** Size separation and characterization of NLPs. (a) Representative size exclusion chromatograms for various NLP preparations as indicated. NLPs were detected by absorption at 280 nm. Typically, fractions comprising 9–13 ml were collected (black horizontal bar). (b) Coomassie-stained SDS PAGE of NLPs. For each preparation, the amount of VAMP2 relative to ApoE was determined using densitometry. (c) Representative negative-stain EM micrographs of NLPs. The top row are SNARE-free NLPs. The bottom row are NLPs loaded with 30 v-SNARE copies. NLPs marked with * are oriented perpendicular to the imaging plane and show the flat disc structure. Scale bar = 25 nm. (d) Distribution of NLP diameters for a representative vNLP15 sample, determined from analysis of micrographs as in (c). A normal distribution fit is shown (red line). (e) Boxplot of representative NLP sizes under various conditions. NLPs containing lipid-anchored VAMP2 (vC45L, vC45M, vC45H for low, medium, and high copy numbers of C45 lipid-anchored VAMP2, bearing ~1, 4, and 15 copies) had sizes comparable to NLPs bearing similar loads of wild-type VAMP2 (vNLP1, vNLP4, and vNLP15). The activity of these NLPs was tested in an established bulk fusion assay with t-SNARE-reconstituted liposomes (*Figure 1—figure supplement 1*).

The following figure supplement is available for figure 1:

**Figure supplement 1.** Bulk content release assay (*Bello et al., 2016*; *Shi et al., 2012*) shows that the fusion of vNLPs with t-SNARE-reconstituted small unilamellar vesicles (t-SUVs) is SNARE-dependent.

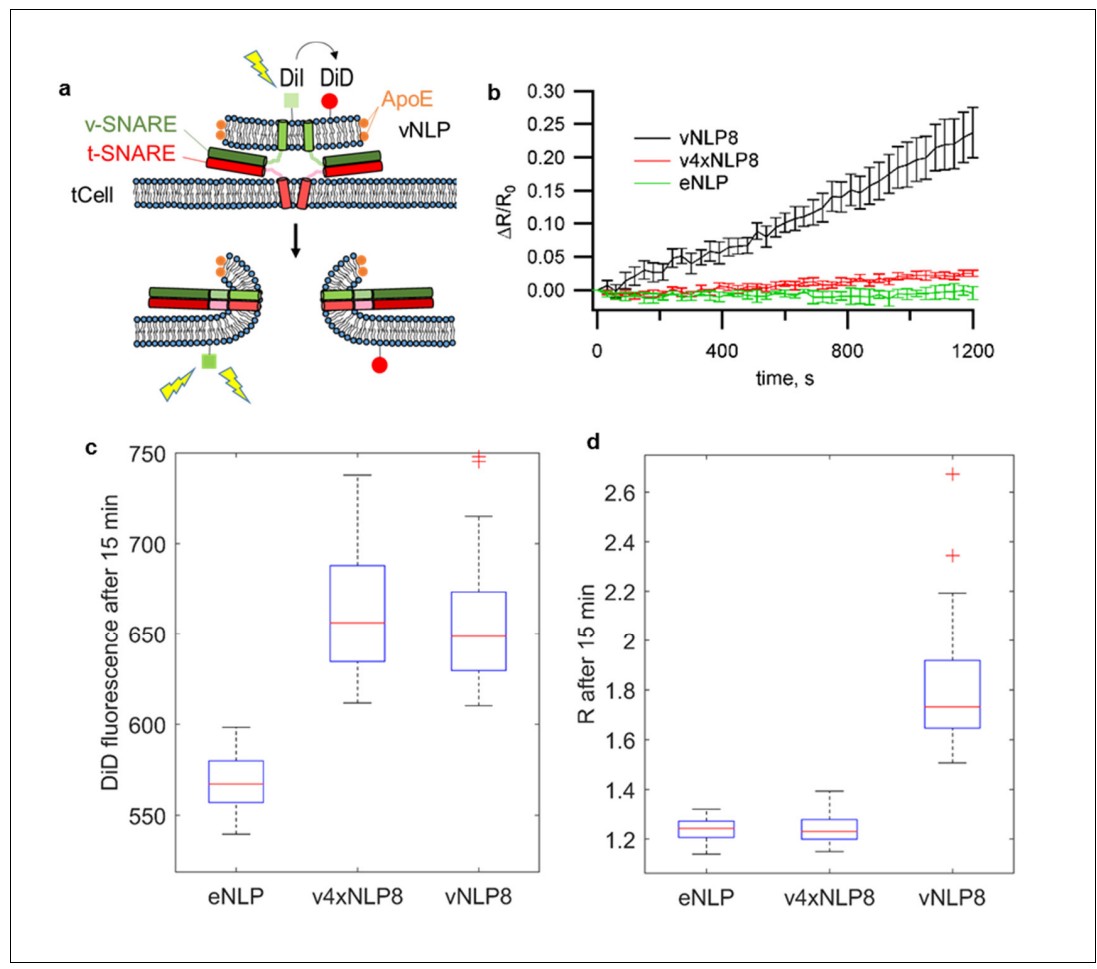

**Figure 2.** vNLPs induce lipid mixing when incubated with flipped t-SNARE cells (tCells). (**a**) Schematic of the assay. (**b**) NLPs co-labeled with one mole % DiI and DiD were incubated with tCells for 30 min at 4°C, a temperature that allows docking but not fusion. Cells were then rinsed with cold PBS to remove free NLPs, and PBS pre-warmed to 37°C was added. Imaging of DiI and DiD fluoresce started shortly after the dish was mounted onto a confocal microscope stage held at 37°C. For each imaging cycle, we sequentially acquired DiI and DiD fluorescence ($\lambda_{ex}$ =561 nm and 647 nm for DiI and DiD, respectively). We quantified cell membrane DiI and DiD fluorescence and calculated the ratio of these two intensities, R. DiI fluorescence reports lipid mixing, while the DiD fluorescence is proportional to the amount of docked NLPs per cell. Thus, the ratio R normalizes the lipid-mixing signal to the amount of docked NLPs. Averages of 69, 73, and 47 regions of interest (± S.E.M.) from 7, 7, and 3 dishes are shown for NLPs loaded with ~eight copies of VAMP2 (vNLP8), for NLPs loaded with ~eight copies of the VAMP2-4x mutant (which are docking-competent but fusion incompetent – v4xNLP8), and for empty NLPs (eNLPs), respectively. (**c,d**) Confocal imaging after 15 min incubation and washing of NLPs with tCells at 37°C. (**c**) DiD fluorescence reflects the amount of docked NLPs per cell. NLPs reconstituted with ~eight copies of VAMP2-4X (v4xNLP8) docked with the same efficiency as wild-type VAMP2 NLPs bearing the same SNARE copy number (vNLP8). (**d**) DiI/DiD fluorescence ratio (R) reports lipid mixing normalized to the amount of docked NLPs per cell. Despite efficient docking, v4xNLP8 did not induce any lipid mixing. eNLP, empty (SNARE-free) NLPs. For (**c**) and (**d**), 6, 10, and 11 dishes were measured and 41, 66, and 63 regions of interest analysed for eNLP, v4xNLP8, and vNLP8, respectively.

The following figure supplement is available for figure 2:

**Figure supplement 1.** Estimation of the extent of lipid mixing.

or with NLPs loaded with eight copies of a v-SNARE construct, VAMP2-4X, carrying mutations in the C-terminal hydrophobic layers (L70D, A74R, A81D, and L84D) (*Figure 2b*). These mutations prevent zippering of the C-terminal half of the SNARE domains (*Wu et al., 2016*; *Krishnakumar et al., 2011, 2013*), a perturbation that does not affect docking (*Figure 2c*) but prevents fusion (*Figure 2b,d*) (*Wu et al., 2016*; *Krishnakumar et al., 2013*). A variation of the assay that avoided the 4°C incubation but prevented live imaging in the presence of the labeled NLPs confirmed these results (*Figure 2d*). We estimate that 4–5% of the docked vNLPs undergo fusion with the flipped t-SNARE cells over the course of ~20 min (*Figure 2—figure supplement 1*). In comparison, fusion between v-SNARE NLPs and t-SNARE liposomes yields a similar extent of lipid mixing over the same period (*Bello et al., 2016*).

Lipid mixing could result from the merging of only the proximal lipid bilayer leaflets of the vNLPs and the tCells. To test whether full fusion occurred, we loaded the cells with Fluo-4, a fluorescent calcium probe, and monitored calcium signals. If full fusion occurred, then calcium influx through the fusion pores connecting vNLP and tCell membranes should increase cytosolic Fluo-4 signals (*Wu et al., 2016*). This was indeed the case for vNLP8 samples, but not for empty NLPs or for NLPs loaded with VAMP2-4X (*Figure 3*). Using this calcium-influx assay, we also assessed whether pores eventually resealed by washing away the free vNLPs after 5 min of incubation. Cellular calcium levels returned to baseline within a few minutes, suggesting that pores eventually resealed (*Figure 3—figure supplement 1*).

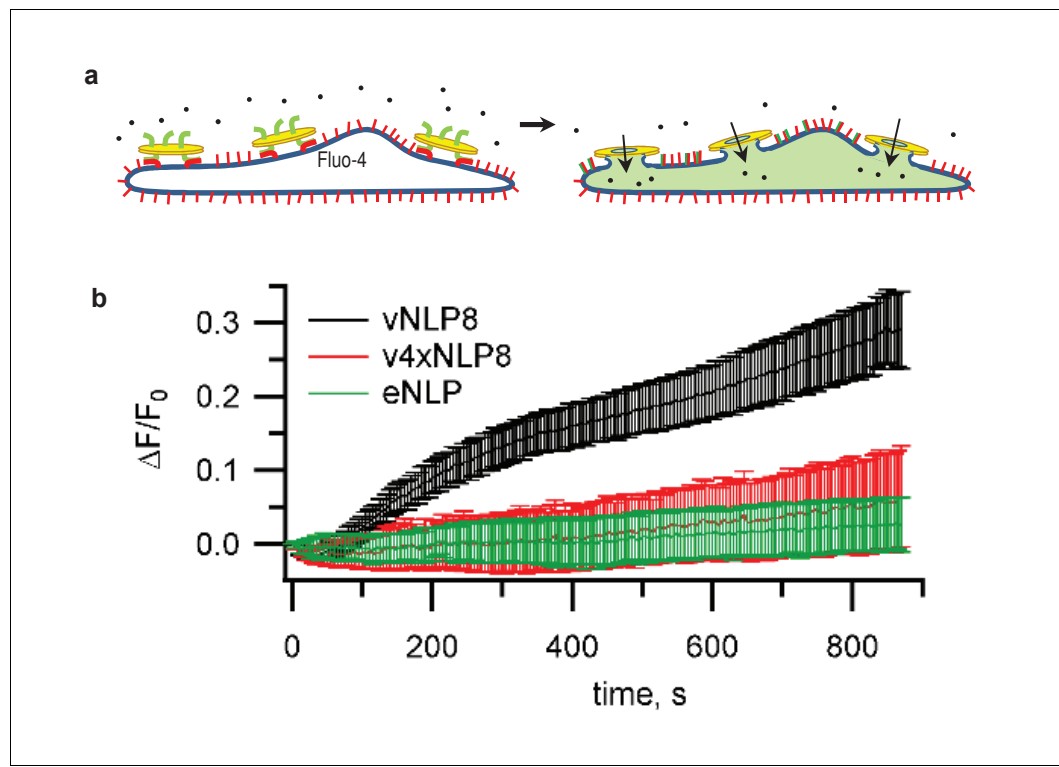

**Figure 3.** Calcium-influx assay. (a) Schematic of the assay. (b) tCells were loaded with the $Ca^{2+}$ indicator Fluo-4, whose fluorescence was imaged as a function of time. Opening of fusion pores allowed $Ca^{2+}$ influx into the cytosol, causing the Fluo-4 signal to increase for vNLP8 (10 dishes), but not for v4xNLP (six dishes) or eNLP (four dishes) samples. The fluorescence from the entire viewfield for each dish was averaged. Displayed errors are S.E. M.

The following figure supplement is available for figure 3:

**Figure supplement 1.** Fusion pores connecting NLPs to cells eventually close.

## Dynamics of single fusion pores

Next, we probed single pores connecting vNLPs to tCell membranes (*Wu et al., 2016*) (*Figure 4*). We voltage-clamped a tCell, in the cell-attached configuration, that was ectopically expressing 'flipped' neuronal/exocytotic t-SNAREs syntaxin1 and SNAP25 (*Hu et al., 2003*) (*Figure 4*). NLPs that were reconstituted with eight copies of the complementary neuronal v-SNARE VAMP2/synaptobrevin (vNLPs) were included in the pipette solution (100 nM vNLPs, 120 µM lipid); they diffused to the pipette tip and fused with the patch. Because an NLP is not a closed structure like a vesicle, its fusion with the voltage-clamped membrane patch establishes a direct conduction pathway between

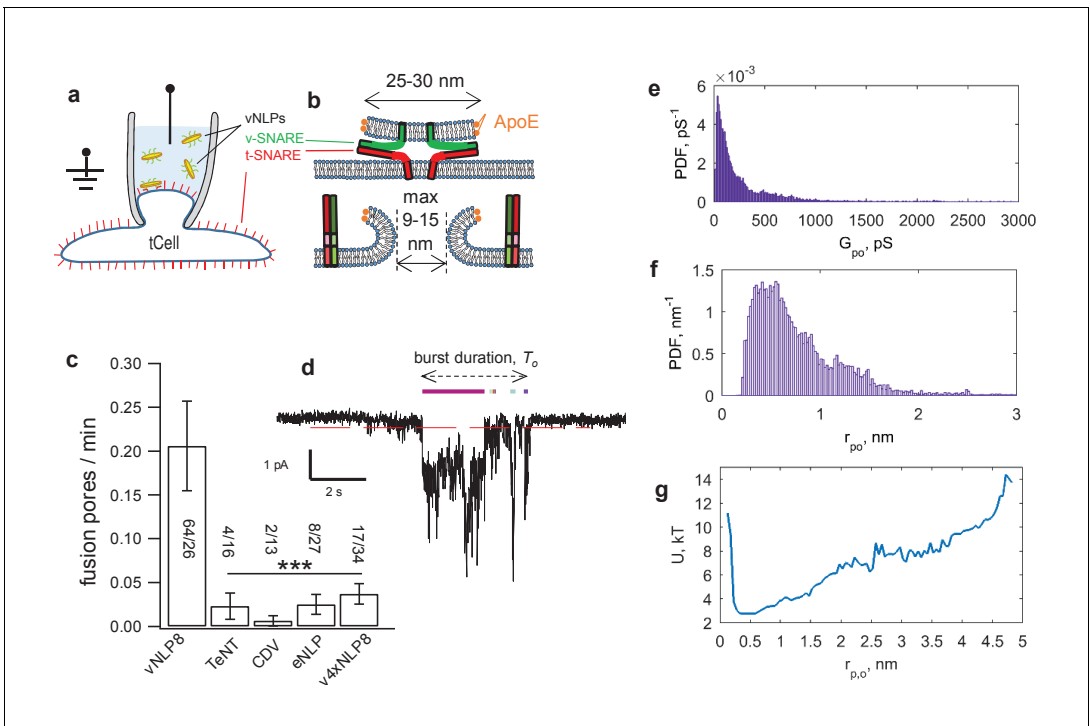

**Figure 4.** Detection of single-pores between vNLP nanodiscs and tCells. (**a, b**) Schematic of the assay. A glass pipette seals a patch on the tCell membrane. The pipette solution includes NLPs. When a vNLP fuses with the tCell membrane (**b**), a nm-sized pore opens and connects the cytosol to the pipette solution. Thus, currents through voltage-clamped pores report fusion and pore properties with sub-ms time resolution. In (**b**), the bilayers, the SNAREs and the NLP are drawn approximately to scale. The light, medium and dark shades of green and red indicate the transmembrane, linker, and SNARE domains of the v- and t-SNAREs, respectively. (**c**) Pores are SNARE-induced. When empty NLPs (eNLPs), the cytoplasmic domain of VAMP2 (CDV), the tetanus neurotoxin light chain (TeNT), or NLPs loaded with the docking-competent, fusion-incompetent VAMP2-4X mutant (v4xNLP8) were used, only a very low level of current activity was recorded compared to the currents resulting from NLPs loaded with ~eight copies of wild-type v-SNAREs. The number of pores/patches are indicated for each condition. (*** indicates p<0.001, t-test against vNLP8). (**d**) An example of a fusion pore current 'burst'. Fusion leads to fluctuating and flickering currents that are well separated in time from one another. A threshold (red dotted line) and a minimum crossing time are imposed to define pore open periods (Materials and methods and *Wu et al., [2016]*). Detected sub-openings are indicated with colored bars above the current trace. (**e**) Average probability density function (PDF) of open-pore conductances. (**f**) Averaged PDF of open-pore radii. Data are from 61 fusion pores, 26 cells. (**g**) Free energy profile calculated from the distribution of pore sizes in (f). Distributions of flicker numbers per pore and burst lifetimes are shown in *Figure 4—figure supplement 1*. Additional examples of current bursts are provided in *Figure 4—figure supplement 2*.

The following figure supplements are available for figure 4:

**Figure supplement 1.** Additional properties of single fusion pores connecting NLPs loaded with eight copies of VAMP2 and flipped t-SNARE cells (64 pores from 26 cells).

**Figure supplement 2.** Additional examples of current bursts.

**Figure supplement 3.** Mycoplasma contamination does not affect fusion with NLPs.

the cytosol and the pipette solution, leading to direct-currents whose magnitude reflects pore geometry (*Wu et al., 2016*).

Currents appeared in bursts with an average frequency of ~0.2 bursts per min, or ~2.5 per patch (*Figure 4c*). The very low burst frequency, together with small unitary conductances (see below), strongly suggest that each burst represents currents passing through a single pore (*Wu et al., 2016*). Currents fluctuated and returned to baseline multiple times, as if the fusion pore fluctuated in size and opened and closed repeatedly, i.e. flickered (*Figure 4d* and Materials and methods). Nucleation was blocked when the cytoplasmic domain of VAMP2 (CDV) was included in the pipette solution, vNLPs were treated with the tetanus neurotoxin light chain (TeNT), or when empty NLPs (eNLPs) were used (*Figure 4c*). CDV competes with the NLP v-SNAREs for binding to the flipped t-SNAREs on the patch surface, and TeNT cleaves VAMP2. When we used VAMP2-4X, pore nucleation rate was not significantly different than for any of the other negative controls (*Figure 4c*). Because this construct allows efficient docking (*Figure 2c*) but is fusion incompetent (*Figure 2b,d*) (*Wu et al., 2016*; *Krishnakumar et al., 2013*), this result indicates that ApoE does not induce pores even when kept in close proximity to the target membrane. Collectively, these observations indicate that, similar to their smaller ND courterparts (*Wu et al., 2016*), NLPs fuse with liposomes or cell membranes in a strictly SNARE-dependent manner.

Combining data from 64 current bursts, we obtained distributions for vNLP8-tCell fusion pores as shown in *Figure 4e–f* and *Figure 4—figure supplement 1*. The number of pore flickers and burst durations were well described by geometric and exponential distributions, respectively, with $N_{\text{flickers}} = 16 \pm 2.7$ flickers per burst and $T_o = 10.3 \pm 2.2$ s (mean ± S.E.M.), as would be expected for discrete transitions between open, transiently blocked, and closed states (*Sakmann and Neher, 2009*) (*Figure 4—figure supplement 1*).

Conductances in the open-state and corresponding radii were broadly distributed (Materials and methods and *Figure 4e,f*), with mean $\langle G_{po} \rangle = 450$ pS (S.E.M. = 169 pS), and $\langle r_{po} \rangle = 0.84$ nm (S.E.M = 0.09 nm), respectively. Surprisingly, these values were significantly less than the maximum possible value predicted from NLP dimensions (*Figure 4b*). This suggested a substantial inherent resistance to pore expansion, independent of the constraints imposed by the NLP dimensions. To quantify the resistance, we computed the apparent pore free energy $U(r_{\text{po}})$ from the distribution of pore radii, $P(r_{\text{po}}) \sim e^{-\frac{U(r_{\text{po}})}{kT}}$. This suggested that ~2 kT energy was required for every 1 nm increase in pore radius above the most likely value $r_{po} \approx 0.5$ nm (*Figure 4f,g*).

## A few SNARE complexes are sufficient to create a fusion pore, but many more are needed to dilate it

We then varied the number of SNAREs, and found that just a few SNARE complexes are sufficient to create a fusion pore, but many more are needed to dilate it. We repeated the measurements shown in *Figure 4* using NLPs loaded with total v-SNARE copy numbers ranging from 1 (vNLP1) to ~30 (vNLP30, ~15 copies per face) (*Figure 5*). Pore nucleation required at least two v-SNAREs per NLP face and maximal nucleation rates were reached at around the same value (*Figure 5a*). By contrast, when ≥four v-SNAREs per NLP face were present, the pore conductances (*Figure 5b*) and radii (*Figure 5—figure supplement 1b*) were significantly larger than the SNARE-free values and increased dramatically as the copy number per NLP face reached 15. Conductance fluctuations about the mean increased even more sharply (*Figure 5—figure supplement 1a*), while burst lifetimes and pore open probability showed a more gradual increase (*Figure 5—figure supplement 1c,d*). Thus, different numbers of SNARE complexes cooperate at the distinct stages of fusion pore nucleation and pore dilation.

Is the increase in the mean pore conductance as the SNARE copy numbers are increased (*Figure 5b*) due to the appearance of multiple small pores per NLP or due to an increase in the mean size of a single pore? The latter is much more likely, for the following reasons. First, a probe that cannot pass through small pores becomes permeant to pores when large copy number vNLPs are employed (*Figure 5—figure supplement 4*). If multiple small pores were present when vNLP30 are used, then the probe should be equally impermeant. The probe employed was N-methyl-D-glucamine (NMDG$^+$), a large ion of ~$1.1 \times 0.5$ nm in size without its hydration shell (*Melikov et al., 2001*), which replaced sodium in the pipette solution. Conductance was low when ~15 nm MSP nanodiscs with eight copies of v-SNAREs (vMSP8) were used (*Wu et al., 2016*), but not affected

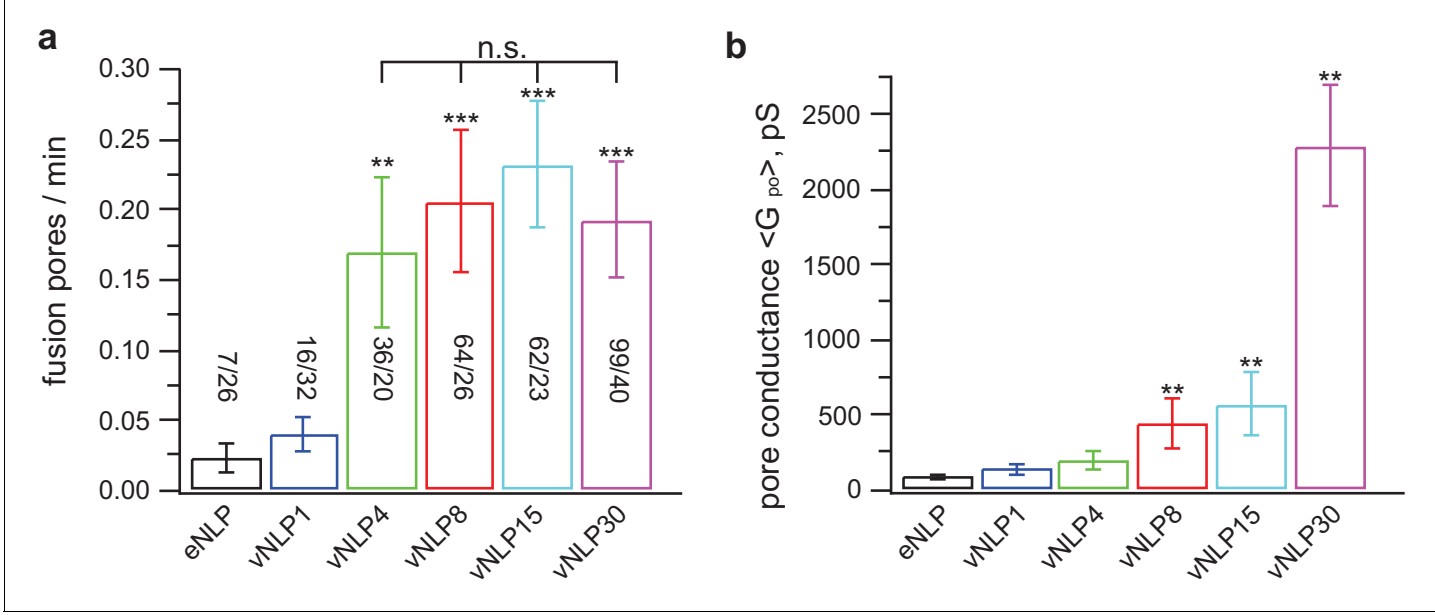

**Figure 5.** Only a few SNARE complexes are required to nucleate a pore, but more than ~15 are required to dilate it. ( a) Pore nucleation rate as a function of total v-SNARE copy number per NLP. Copy numbers per NLP face are approximately 0, 1, 2, 4, 7.5, and 15 for eNLP and vNLP1 through vNLP30, respectively. Pore nucleation requires ~two copies per NLP face and saturates at two to four copies per NLP face. n.s. indicates no statistically significant differences for the mean fusion rates among vNLP4, vNLP8, vNLP15, and vNLP30 samples, as assessed by an analysis of variance (ANOVA) and multiple pairwise comparisons of the group means. The source and analysis files are provided as *Figure 5—source data 1*. (b) Mean single-pore conductance, $G_{po}$ as a function of number of v-SNARE copies loaded into NLPs. $G_{po}$ increases rapidly as increasing numbers of v-SNAREs are loaded per NLP. At the maximum value tested, ~15 copies per NLP face, $G_{po}$ is far from saturating. The number of pores analyzed/total number of cells is indicated for each condition in (a). **, *** indicate $p<0.01$ and 0.001, respectively, using the two-sample t-test (a) or the Kolmogorov-Smirnov test (b) to compare with eNLP. Additional pore properties are shown in *Figure 5—figure supplement 1*. Properties of pores induced using lipid-anchored v-SNAREs are shown in *Figure 5—figure supplement 2*.

The following source data and figure supplements are available for figure 5:

**Source data 1.** Statistical analysis of fusion rates reported in *Figure 5a*.
**Figure supplement 1.** Additional pore properties as a function of v-SNARE copy number per NLP.
**Figure supplement 2.** Larger numbers of lipid-anchored v-SNAREs promote pore dilation.
**Figure supplement 3.** Swapping the locations of the v- and t-SNAREs does not affect pore properties.
**Figure supplement 4.** Permeability of pores to NMDG$^{+}$.

when ~23 nm NLPs bearing 30 v-SNAREs were employed (vNLP30). These results are consistent with those of *Bello et al. (2016)*, who showed that progressively larger cargo could be released from t-SNARE liposomes during fusion with vNLPs as the v-SNARE copies per NLP was increased. Second, conductance of $n$ small pores in a single NLP would be additive, giving total conductance equal to $G_{po} = n \times g_{po}$, where $g_{po}$ is the mean open-pore conductance of a small pore. Doubling the SNARE copies would presumably at most double $n$, and by consequence, total conductance. The fact that we find faster than linear increase in mean pore conductance (*Figure 5b*) is consistent with each NLP bearing a single pore whose size increases with increasing SNARE copies. Third, if multiple small pores occurred per NLP, this should be evident in the distribution of point-by-point conductance values, with peaks at $n \times g_{po}$, where $n = 1, 2, 3 \ldots$. Instead, for the distribution of mean $G_{po}$ for vNLP30 we find a peak at ~300 pS, and a broad peak at ~3–14 nS (*Figure 6b*). If the typical small pore has 300 pS conductance, then to have ~6 nS (typical large conductance), there would have to be ~20 small pores per NLP. It is hard to imagine that this many pores could coexist in this small

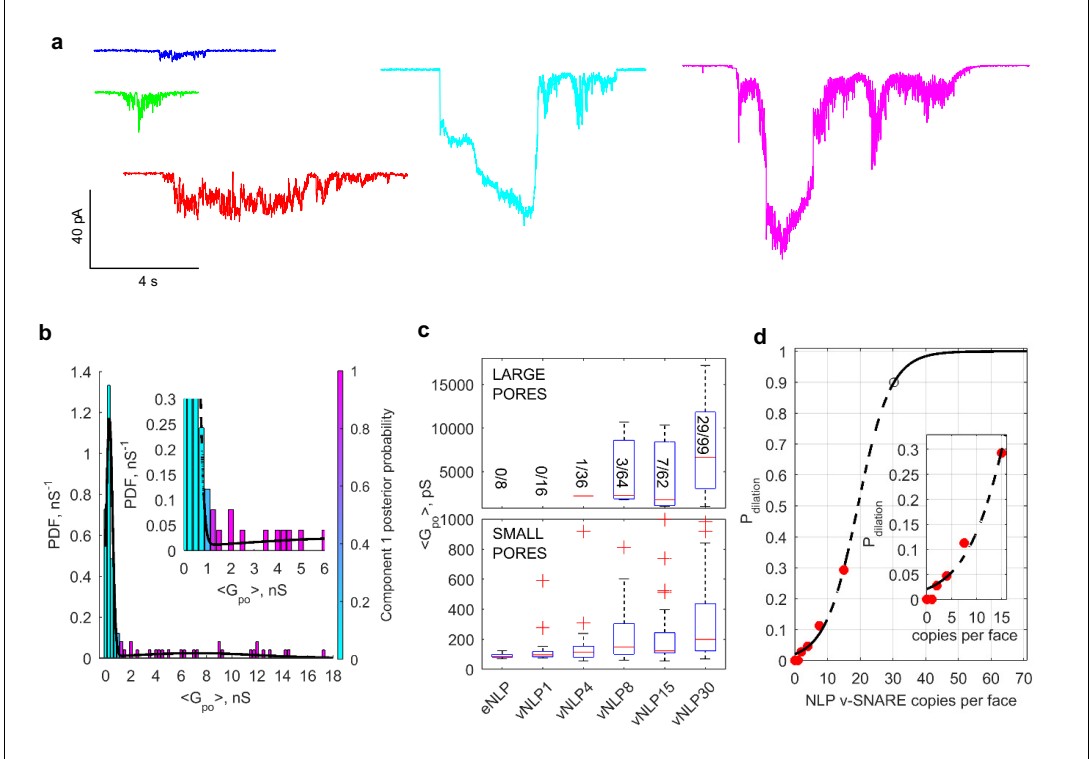

**Figure 6.** Increasing v-SNARE copy numbers increases the occurrence of large pores. (a) At low copy numbers, all pores produced small amplitude currents (leftmost traces). As copy numbers increased, most pores still produced small-amplitude currents, but an increasing fraction had much larger currents, such as those shown in the two traces on the right. (b) The probability density function of mean open-pore conductance values $G_{po}$ from 99 vNLP30-tCell fusion pores was fitted with a Gaussian mixture model with two components. The data clustered into two Gaussian distributions centered around 300 pS and 7.21 nS, separated at ~1 nS. For every bin, the probability of belonging to component one is color-coded with the color map indicated to the right of the plot. The inset shows a zoom to the transition region between the two components. (c) Individual pores were classified as low ($G_{po}<1$ nS) or high ($G_{po}>1$ nS) conductance. The distribution of mean conductances is shown as a series of box plots for the v-SNARE copy numbers tested. The number of large pores/total number of pores is indicated for each group. (d) Probability of pore dilation, $P_{dilation}$, defined as the fraction of pores in the high-conductance category in (c) as a function of SNARE copy number per NLP face (red dots). The dashed line is a fit $P_{dil} = \exp\left(\frac{N_{SNARE}-N_o}{b}\right) / \left(1 + \exp\left(\frac{N_{SNARE}-N_o}{b}\right)\right)$, where $N_o$ is the copy number at which $P_{dil} = 0.5$, and $b$ measures the width of the transition. Best fit parameters were (with 95% confidence intervals) $N_o = 19.3$ (16.9, 21.7), and $b = 5.0$ (3.3, 6.7) ($R^2$: 0.97). The black open circle indicates that $P_{dil} = 0.9$ requires 30 SNAREs. See *Figure 6—figure supplement 1* for a plot of open-pore conductance fluctuations relative to mean as a function of mean open-pore conductance.

The following figure supplement is available for figure 6:

**Figure supplement 1.** Open-pore conductance fluctuations relative to mean, $\langle G_{po}^2 \rangle / \langle G_{po} \rangle$, as a function of mean open-pore conductance, $\langle G_{po} \rangle$.

area. Finally, unless the multiple pores occurred simultaneously, we would also find that the fusion rate increases with copy number. Instead, the rate saturates at around two copies (*Figure 5a*). In conclusion, although we cannot rule out that, very occasionally, a small number of pores may simultaneously appear in a single NLP, all the evidence suggests that this cannot be very common.

Previous reports suggested that pore nucleation is promoted by the assembly of the v- and t-SNARE transmembrane domains (TMDs) (*Wu et al., 2016*; *Shi et al., 2012*; *Fdez et al., 2010*). To test whether pore dilation also required the TMDs, we replaced the v-SNARE TMDs with lipid anchors (*Shi et al., 2012*). We used long-chain anchors that span both leaflets of the bilayer because previous work suggested that lipid anchors spanning a single leaflet are not efficient in inducing full fusion (*McNew et al., 2000*; *Chang et al., 2016*). Lipid-anchoring VAMP2 into NLPs significantly reduced pore nucleation frequency and increased the mean burst duration (*Figure 5—figure supplement 2a,b,e*), consistent with previous work using smaller MSP NDs (*Wu et al., 2016*)

and with reduced overall fusion efficiency reported for lipid-anchored VAMP2 (*Shi et al., 2012*; *Chang et al., 2016*). Importantly however, fusion pores induced by lipid-anchored v-SNAREs displayed the same trends as their intact counterparts: as VAMP2-C45 copy number increased, so did mean conductance, fluctuations, burst lifetimes, and pore radii, but the pore open probability during a burst varied little (*Figure 5—figure supplement 2*). These results suggest that specific interactions between v-SNARE and t-SNARE TMDs are not critical for cooperative pore dilation by SNARE proteins.

The target membrane during exocytosis (where the t-SNAREs reside) is the inner leaflet of the plasma membrane, which is rich in acidic phospholipids. By contrast, in our vNLP-tCell fusion assay, the target membrane is the outer leaflet of the plasma membrane, which is largely devoid of negatively charged lipids. In general, a limitation of our system is that the lipid composition of the outer leaflet of the t-SNARE-presenting cell differs substantially from that of the plasma membrane inner leaflet, and lipid composition can play a key role in fusion. To test whether the target membrane composition affected fusion, we swapped the locations of the v- and t-SNAREs and fused flipped v-SNARE cells with t-SNARE NLPs. This allowed us to have a better mimic of the inner plasma membrane leaflet composition on the target membrane (now the tNLP membrane). This swap resulted in similar fusion rates and pore properties for two different SNARE copy numbers per NLP (*Figure 5—figure supplement 3*), suggesting that fusion mediated by SNAREs alone may not be very sensitive to target membrane composition within a certain range (*Stratton et al., 2016*).

Interestingly, the increase in mean pore conductance as v-SNARE copy numbers are increased does not occur homogeneously across all pores. For vNLPs bearing four or more total copies of v-SNAREs, we found two types of fusion pores. Most had small mean conductance $\leq 1$ nS, but with increasing SNARE load, an increasing fraction of pores had much larger conductances of a few nS (*Figure 6a*). By contrast, for NLPs that contained no copies or just one copy of VAMP2, the pores that occasionally occurred all had small mean conductance $\leq 1$ nS. The distribution of mean conductances for individual pores revealed two components for vNLP30 (*Figure 6b*), with a sharp peak at ~300 pS (71% of total mass) and a much broader population centered at 7.21 nS, separated at ~1 nS. Conductance fluctuations increased sharply for larger pores with $\langle G_{po} \rangle \geq 1$ nS (*Figure 6—figure supplement 1*), indicating a change in behavior above this threshold. Thus, multiple criteria indicated ~1 nS as a cut-off that separated small and large pores. We applied this cut-off to all NLPs tested, and clustered pore conductances in each NLP group into low ($\langle G_{po} \rangle < 1$ nS) and high conductance ($\langle G_{po} \rangle > 1$ nS) states (*Figure 6c*). The occurrence of high conductance pores increased with increasing SNARE copy number, suggesting that dilation of the pores to >1 nS (corresponding to $r_{po} \approx 1.7$ nm) is facilitated by SNARE crowding.

We defined the pore dilation probability, $P_{dilation}$, as the fraction of pores in the high-conductance state for a given SNARE copy number (*Figure 6c*), and plotted $P_{dilation}$ as a function of v-SNARE copies per face (*Figure 6d*). Even at the maximum SNARE load of ~15 copies per face, $P_{dilation}$ was ~0.30, far from saturating. To estimate how many SNARE complexes would be required to reach saturation, we assumed that the ratio between the probabilities of high and low conductance states, $P_{dilation}/(1 - P_{dilation})$ is equal to a Boltzmann factor $e^{-\Delta E/kT}$, with $\Delta E$ the difference between the energy levels of the two conductance states, and $kT$ thermal energy. Making the simplest assumption that $\Delta E \propto N_{SNARE} - N_0$, where $N_{SNARE}$ is the number of SNARE complexes involved and $N_0$ is the copy number that would make $P_{dilation} = 0.5$, we found $N_0 = 19.3$ (*Figure 6d*). Thus, pore dilation with $P_{dilation} = 0.90$ would require $N \approx 30$ complexes (*Figure 6d*, open black circle), about 10-fold more than is required for nucleation (*Figure 5a*).

## SNARE crowding generates entropic forces that drive pore expansion

The characteristic fusion pore free-energy function $U(r_{po})$ progressively softened as the SNARE copy number increased (*Figure 7a*). The minimum at $r_{po} \approx 0.5$ nm did not shift, but for larger pore radii, the slope decreased and the profile broadened. These free-energy profiles quantify how fusion-pore dilation is driven by SNARE proteins. For example, an energy ~6 $kT$ is required to expand the one-SNARE fusion pore from its preferred radius of ~0.5 nm to a 3-fold larger pore, showing that such an expansion is unlikely to occur spontaneously. On the other hand, with four SNAREs per face, the same expansion requires only ~3 $kT$, and only ~2 $kT$ with 15 SNAREs (*Figure 7a*), bringing the expansion within reach of spontaneous fluctuations. The broad and shallow profile suggests that a

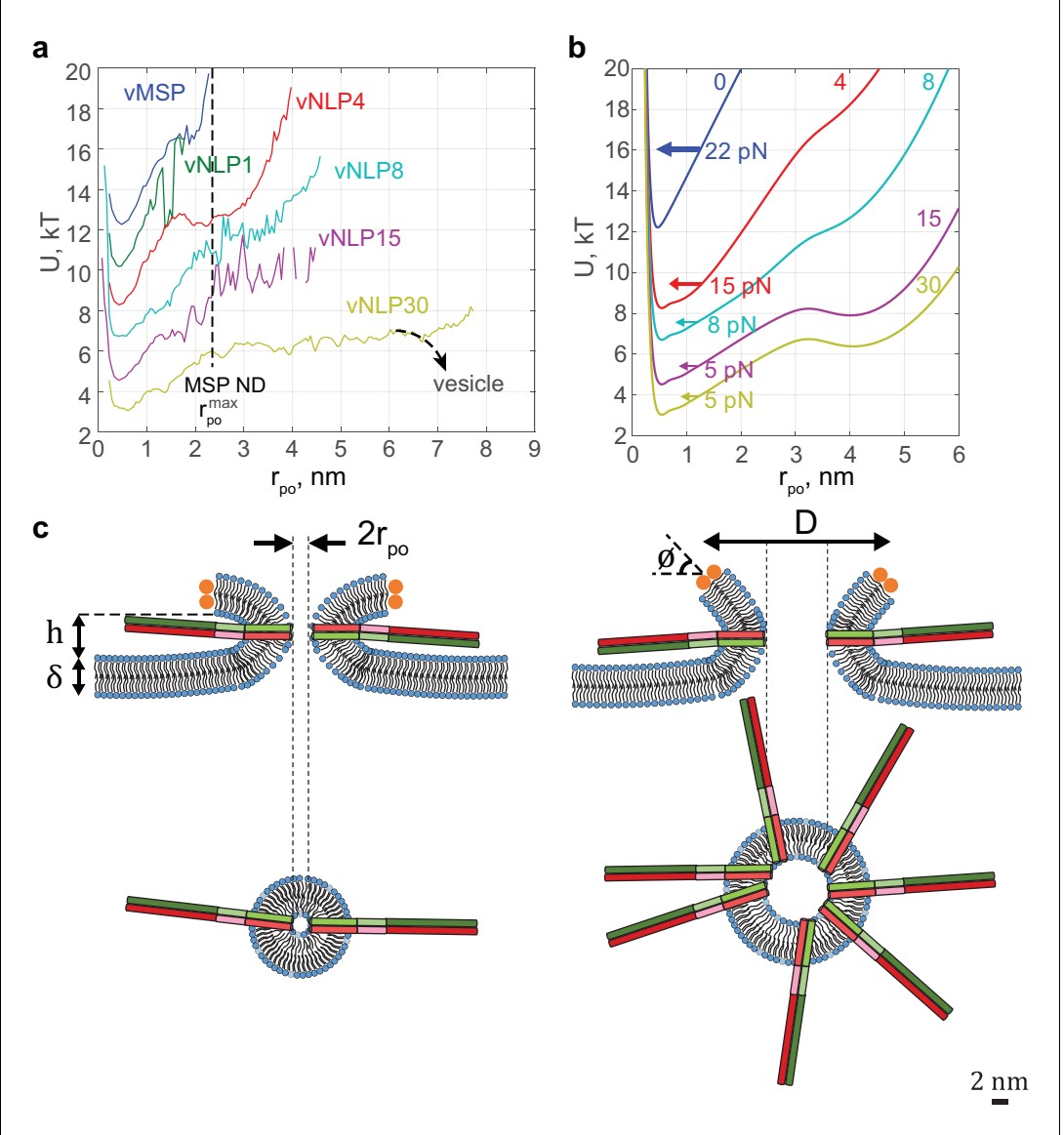

**Figure 7.** Free energy profiles for pore dilation, experimental results and model predictions. The mathematical model describes a mechanism of pore expansion in which SNARE-crowding generates entropic expansion forces. (a) Open-pore free-energy landscape $U(r_{po})$ for different SNARE copy numbers. Increasing SNARE copy numbers in NLP discs softens the energy barrier against pore expansion. For vNLP30 discs, the profile starts rising above $r_{po} \approx 7$ nm (expected maximum size $r_{po,\,max}^{NLP} \approx 7 - 8$ nm). If a vesicle were fusing instead of a NLP, dilation would presumably relax pore curvature and lower the energy (dashed curve marked 'vesicle'). vMSP data were obtained in earlier work (**Wu et al., 2016**) using smaller, ~16 nm diameter nanodiscs stabilized by the membrane scaffold protein (MSP), with 7–9 v-SNARE copies. The maximum allowable pore size is limited to slightly above the 2 nm radius of MSP discs. The same energy minimum around $r_{po} \approx 0.5$ nm is found regardless of copy numbers or the size of disc used, suggesting that this minimum represents an inherent property of fusing bilayers. (b) Corresponding free-energy profiles predicted by a mathematical model of the fusion pore with SNAREs (Materials and methods, **Figure 7—figure supplement 1**, and **Table 1**). Each curve shows the copy number and the net inward force (averaged over all pore sizes $r_{po} > 1.5$ nm) tending to close down the pore to the minimum energy value. Membrane bending and tension resist pore expansion with a total force ~22 pN (SNARE-free pore, blue). In the presence of SNAREs, crowding effects produce an expansive entropic force that reduces the net inward force. The net force is progressively lowered with increasing numbers of SNAREs, reaching ~5 pN with 15 SNAREs. (c) Schematic illustrating a proposed SNARE-mediated pore expansion mechanism. Left: a few SNAREs can nucleate a pore, but dilation beyond a few nm is unlikely. Right: with many SNAREs, crowding generates expansion forces that are sufficient to offset the intrinsic bilayer resistance and to expand the pore. $h, \delta, D$ and $\phi$ are the height of

*Figure 7 continued on next page*

*Figure 7 continued*

the pore, the thickness of the membrane, the mean diameter of the vNLP discs, and the angle of twisting of the ApoE proteins, respectively. For definitions of other model parameters, see Materials and methods.

The following figure supplement is available for figure 7:

**Figure supplement 1.** Results of the mathematical model of the fusion pore in the presence of SNAREpins.

fully developed pore would then result, if the NLP were replaced by a vesicle as in the physiological setting (*Figure 7a*). These effects can equivalently be phrased in terms of force: ~24 pN opposes pore expansion with one SNARE, but this is lowered to ~8 pN and ~5 pN by 4 and 15 SNAREs per face, respectively.

To help to elucidate the molecular mechanism underlying SNARE-mediated pore dilation, we developed a coarse-grained mathematical model that assumed that the bilayer-SNARE system is equilibrated, consistent with the long-lived current bursts, so that $U(r_{po})$ is then the true thermodynamic free energy (see Materials and methods and *Table 1* for model parameters). In our model, for a given pore size, the free energy represents an average over possible fusion pore heights, pore shapes and SNARE complex configurations. SNAREs can be fully zippered at the fusion pore waist, or they may unzipper and roam (*Figure 7—figure supplement 1*).

The model free energies reproduced experimental profiles with physiologically realistic parameters (*Figure 7b*). Protein-free pores resisted expansion because a bigger pore has greater area: a 1 nm increase in pore radius required ~3.0 *kT* work against membrane tension and increased membrane bending energy by ~2.4 *kT*. Thus, a net force ~22 pN resists pore expansion, close to the experimental value of ~24 pN (*Figure 7a*). When SNAREs were present, zippering of SNARE linker domains (*Gao et al., 2012*) and TMDs drove several SNAREs to fully assemble at the fusion pore waist, where crowding generated an entropic pore expansion force (*Figure 7c*). Bigger pores were associated with more zippered SNAREs at the waist (*Figure 7—figure supplement 1*). With 15 SNAREs per face, the entropic expansion force of 17 pN was within 5 pN of the 22 pN resistance.

**Table 1.** Parameters used in the analytical model of fusion pores. (A) Measured by fitting a cylinder to the part of the fully zippered SNARE protein structure without the TMDs, produced using PyMOL software with PDB code 3HD7 (http://www.pymol.org). (B) Estimated in this study as a fitting parameter. (C) Measured by *Mitra et al. (2004)*. (D) Measured in this study (*Figure 1*). (E) Calculated using a weighed average of the $P_0$ of palmitoyl-2-oleoyl phosphatidylcholine (POPC) and 1,2-dioleoyl phosphatidylserine (DOPS) from *Rand and Parsegian (1989)*. The weighed average of the two pressure parameters, according to the 85:15 molar ratio of POPC:DOPS present in the NLPs in this study, is used to obtain $P_0$. We assumed that the hydration properties of POPC are the same as those of 1,2-dioleoyl phosphatidylcholine (DOPC). (F) Values of $\kappa$ in previous studies range from 10–50 kT (*Cohen and Melikyan, 2004*; *Brochard and Lennon, 1975*; *Khelashvili et al., 2013*; *Marsh, 2006*). A commonly used value in studies is $\kappa = 20\text{kT}$ (*Jackson, 2009*), which we used here. (G) Calculated on the basis of an atomistic molecular dynamics study of the t-SNARE TMD, which shows that these domains explore angles of ~$10^0$ around their equilibrium position in a bilayer (*Knecht and Grubmüller, 2003*).

| Symbol | Meaning | Value | Legend |
|---|---|---|---|
| b | Thickness of SNARE bundle | 2 nm | (A) |
| $\varepsilon_{\text{zip}}$ | Energy of zippering of v- and t-SNAREs | 9.6 kT | (B) |
| $\delta$ | Thickness of the plasma membrane | 5 nm | (C) |
| D | NLP diameter | 24 nm | (D) |
| $\lambda$ | Decay length for inter-membrane steric-hydration force | 0.10 nm | (B) |
| $P_0$ | Pressure prefactor for inter-membrane steric-hydration force | $5.0 \times 10^{11}$ dyn/cm$^2$ | (E) |
| $\tau$ | Torque per unit length to twist the ApoE proteins at the NLP boundary | 8.43 pN | (B) |
| $\kappa$ | Bending modulus of the lipid bilayer | 20 kT | (F) |
| $\gamma$ | Membrane tension | 0.66 pN nm$^{-1}$ | (B) |
| $\Omega_z$ | Solid angle explored by bending of zippered SNAREs | 0.05 sr | (G) |

Consistent with our experiments using lipid-anchored v-SNAREs, when we ran the model with lowered excluded volume among zippered SNAREs to mimic the lipid anchor, the best fit total zippering energy was reduced by $<1\ kT$ (see Materials and methods). This suggests that the driving force for SNARE zippering that underlies pore expansion does not rely on putative v- and t-SNARE TMD interactions, but has a significant contribution from non-specific interactions that favor the alignment of membrane anchors.

## Discussion

In summary, we find that a few SNAREs can nucleate a fusion pore, consistent with previous findings (*Bao et al., 2016*; *Shi et al., 2012*; *Sinha et al., 2011*; *Mohrmann et al., 2010*; *van den Bogaart et al., 2010*; *Karatekin et al., 2010*), but the pore is highly unlikely to expand significantly without additional forces. Pore dilation is resisted by intrinsic bilayer properties (*Chanturiya et al., 1997*; *Chizmadzhev et al., 1995*; *Jackson, 2009*), but promoted by the action of many SNAREs that cooperatively exert expansion forces of entropic origin (*Figure 7c*).

In our study, pores fluctuated in size, and closed and opened (flickered) multiple times before resealing, as do exocytotic fusion pores that have been recorded from neuroendocrine cells or neurons (*Lindau et al., 2003*; *Staal et al., 2004*; *Fulop et al., 2005*; *He et al., 2006*; *Klyachko and Jackson, 2002*). Each such burst lasted several seconds on average. Confinement of the pore radius by the NLP scaffold to $\leq$7–8 nm probably contributed to this long lifetime. However, we suggest the lifetime also reflects the natural tendency of small pores to resist dilation, for the following reasons. First, the most likely pore radius, ~0.5 nm (*Figures 4f* and *7a*), is much smaller than the maximum allowed by the NLP geometry, ~7–8 nm. Second, fusion pores connecting protein-free bilayers flicker for seconds, and do not dilate unless increased membrane tension is applied (*Chanturiya et al., 1997*). Third, pores lasting of the order of a second or longer have been documented during exocytosis using capacitance recordings made, for example, in beta cells secreting insulin (*Hanna et al., 2009*; *MacDonald et al., 2006*) or during synaptic vesicle fusion (*He et al., 2006*). Amperometry often reports shorter pore lifetimes when compared to capacitance measurements (*Chang et al., 2015*); it may underestimate pore lifetimes because no signal can be observed once all cargo is released. It is also possible that a pore can reseal after partial dilation giving rise to an amperometric spike, leading to an underestimation of pore lifetime based on the pre-spike foot feature alone (*Mellander et al., 2012*). Fourth, a recent FRET-based study suggested the existence of long-lived, narrow fusion pores during neuronal SNARE-driven fusion between surface-tethered liposomes, dilation of which was promoted by Synaptotagmin-Ca$^{2+}$ and Complexin (*Lai et al., 2013*). Finally, various theoretical models suggested that small pores are metastable (*Chizmadzhev et al., 1995*; *Jackson, 2009*; *Nanavati et al., 1992*).

We measured a low fusion efficiency in our assay (4–5% of docked NLPs undergo lipid mixing within ~20 min (*Figure 2—figure supplement 1*), possibly due to the absence of factors known to be essential for exocytosis in our assay. Indeed, the low fusion efficiency in SNARE-only reconstitutions that lack other factors that are required in vivo for exocytosis, such as Munc13, is well documented (*Bao et al., 2016*; *Bello et al., 2016*; *Weber et al., 1998*; *Liu et al., 2016*; *Hernandez et al., 2014*; *Diao et al., 2013*). Another possibility is that the NLPs that actually fuse may be biased toward higher copy numbers of proteins, if higher copy number corresponded to higher fusion rates. While we cannot categorically exclude this possibility, we think it unlikely. First, the fusion rates that we measured were statistically indistinguishable for copy numbers $\geq$four (vNLP4, vNLP8, vNLP15 and vNLP30, *Figure 5a*). Thus, there is no evidence of a bias due to differential fusion rates. Second, even if there were such a bias, our results would still correctly report the general trend of pore properties versus copy number, as can be seen from the following argument. Assume a Poisson distribution for the copy number up to a maximum of 30, the maximum attainable value in our experiments (presumably a packing constraint). Then, for large mean copy numbers, this distribution has a small width, so that even if the ~5% fused fraction corresponds to the tail of this distribution, the copy numbers involved will not be much greater than the mean value. Thus, the typical copy number of the NLPs whose pore properties are measured would still be an increasing function of the mean value.

A wide range of SNARE copy number requirements for fusion have been reported in the literature (*Bao et al., 2016*; *Shi et al., 2012*; *Sinha et al., 2011*; *Mohrmann et al., 2010*; *van den*

*Bogaart et al., 2010*; *Karatekin et al., 2010*; *Hernandez et al., 2014*; *Domanska et al., 2010*; *Montecucco et al., 2005*), depending on the system studied and the read-out used for fusion. Most studies concluded that only a few copies of neuronal SNAREs are sufficient for calcium-triggered exocytosis and fusion of small liposomes (*Bao et al., 2016*; *Shi et al., 2012*; *Sinha et al., 2011*; *Mohrmann et al., 2010*; *van den Bogaart et al., 2010*; *Karatekin et al., 2010*; *Domanska et al., 2010*). Despite this, the average synaptic vesicle carries 70 v-SNARE copies (*Takamori et al., 2006*) and at least as many t-SNAREs are clustered at plasma membrane docking and fusion sites in neuro-endocrine cells (*Knowles et al., 2010*). Our results provide a rationalization for this situation, as they suggest that reliable pore dilation may require the engagement of many SNARE complexes. As the demands for SNARE cooperativity may be different at different stages of the fusion reaction, inter-pretation of copy number requirements should be made with caution. Methods that rely on lipid mixing or on the exchange of small ions through pores (e.g. capacitance or pH sensing) may mea-sure the requirement for the opening of small fusion pores, which may differ substantially from the requirements for pore dilation.

The action of SNAREs is highly regulated by other proteins during neurotransmitter or hormone release (*Südhof and Rothman, 2009*). In addition to manipulations of SNAREs (*Han et al., 2004*; *Fang et al., 2008*; *Kesavan et al., 2007*), mutations in Munc18 (*Jorgacevski et al., 2011*), Synapto-tagmin (*Wang et al., 2001*, *2003a*, *2003b*), and complexin (*Dhara et al., 2014*) affect fusion pore properties, linking these proteins to pore dynamics. Thus, one must be cautious when extending our SNARE-only results to physiological neurotransmitter and hormone release. Nevertheless, the con-cept of the promotion of pore dilation by protein crowding is a general principle that may hold qual-itatively in the presence of additional components of the physiological fusion machinery. Indeed, a previous study indicated that the availability of SNAREs affects neurotransmitter release kinetics in neurons (*Acuna et al., 2014*), while another suggested release occurred faster at sites with more t-SNAREs (*Zhao et al., 2013*). Thus, we tentatively suggest that some proteins may exert their exo-cytotic regulatory function by organizing SNARE complexes around the fusion site and thereby con-trolling the number that participate, or by sequestering SNAREs to limit that number. Given the steep dilation probability curve (*Figure 6d*), our results suggest a high sensitivity in the balance between transient *versus* full fusion.

## Materials and methods

### Stable flipped SNARE and wild-type HeLa cell culture

HeLa cell lines stably co-expressing flipped v-SNAREs (flipped VAMP2 and cytosolic DsRed2-nes, 'vCells') and t-SNAREs (flipped Syntaxin-1 and flipped SNAP-25 and the nuclear fluorescent marker CFP-nls, 'tCells') were generated in the Rothman laboratory as described (*Giraudo et al., 2006*). The cells were a generous gift from the Rothman laboratory. The cells were maintained in DMEM (4500 mg/L glucose, L-glutamine, sodium pyruvate and sodium bicarbonate) and 10% (v/v) fetal calf serum at 37°C. A new aliquot of cryopreserved cells was thawed after at most three weeks of cell culture and cultured at least five days before data acquisition.

The flipped t-SNARE HeLa cells were tested by PCR Mycoplasma Test Kit I/C (cat. No. PK-CA91-1048, Promo Kine, Heidelberg, Germany), which showed contamination (*Figure 4—figure supple-ment 3A*). Because our assay fuses discs to the surface of these cells, no effect of mycoplasma con-tamination is expected. Indeed the controls in *Figures 2–4* indicate that fusion is SNARE-driven. Nevertheless, we tested any possible impact of mycoplasma contamination on our results by repeat-ing some of our single-pore measurements with cells treated with an antimycoplasma reagent (Plas-mocin, cat. code ant-mpt, InvivoGen, California, USA). Fusion rates and pore properties were indistinguishable when untreated or treated cells were used (*Figure 4—figure supplement 3, B–E*), suggesting that mycoplasma contamination does not affect fusion with NLPs.

### Plasmids, protein expression and purification

Expression and purification of the t-SNARE complex used in vND-tSUV fusion experiments is described in *Parlati et al. (1999)*. The cytoplasmic domain of VAMP2 (CDV) was produced using a method that is similar to an earlier protocol (*Weber et al., 1998*), except that a SUMO vector was used. We followed the methods of *Giraudo et al. (2006)* for expression and purification of the

tetanus neurotoxin light chain, TeNT. VAMP2 proteins were expressed and purified as described earlier by *Shi et al. (2012)*. To produce full-length WT VAMP2, we used the plasmid pET-SUMO-VAMP2 (*Shi et al., 2012*). To produce lipid-anchored VAMP2, we followed the methods of *McNew et al. (2000)* and *Shi et al. (2012)*. We first used a previously described construct (*McNew et al., 2000*) to produce VAMP$^{95}$Cys containing the entire cytoplasmic domain of VAMP2 (residues 1–95) with a C-terminal cysteine residue. We then coupled this protein to maleimidopropionic acid solanesyl ester (maleimide-C45), produced as previously described (*Shi et al., 2012*). For producing MSP NDs, we used the vector pET28-MSP1E3D1 (Addgene, Cambridge, MA) to express and purify MSP1E3D1 as described previously (*Ritchie et al., 2009*), except that we cleaved the MSP proteins directly off the column by TEV protease overnight at 4 °C (*Wu et al., 2016*).

Plasmid pET32a-Trx-His6X-ApoE422K, which we used to express the N-terminal 22 kDa fragment of apolipoprotein E4 (residues 1–199), was kindly provided by Dr Nicholas Fischer, Lawrence Livermore National Laboratory, CA (*Morrow et al., 1999*; *Blanchette et al., 2008*). ApoE422K was expressed and purified as previously described (*Morrow et al., 1999*) with the following modifications. The His6-ApoE422K was cleaved off the Ni-NTA beads (Qiagen, Germantown, MD) using 100U of Thrombin at 4°C overnight. The protein was eluted in 25 mM HEPES, 140 KCl, pH 7.4 buffer containing 1% octylglucoside (OG), and was functional for up to 4 weeks when stored at 4°C. Protein concentrations were determined using the Bradford assay (Bio-Rad, Hercules, CA) with bovine serum albumin as standard.

## Characterization of nanolipoprotein particles (NLP)

Details are provided by *Bello et al. (2016)*. Briefly, nanolipoprotein particles containing VAMP2 (vNLP) were produced using a modified version of the established protocol to generate SNARE-nanodiscs (*Shi et al., 2012*, *2013*). A palmitoyl-2-oleoylphosphatidylcholine (POPC): 1,2-dioleoyl phosphatidylserine (DOPS) = 85:15 molar ratio lipid mixture (Avanti Polar Lipids, Alabaster, AL) was dried under nitrogen flow, followed by vacuum for 1 hr. The lipid film was re-suspended in 25 mM HEPES, pH 7.4, 140 mM KCl, buffer with 1% OG supplemented by the desired amount of VAMP2. The mixture was vortexed at room temperature (RT) for 1 hr followed by the addition of ApoE422K and vortexed for another hour at RT. The ApoE422K:VAMP2:lipid ratio was varied to tune the v-SNARE copy number per NLP as 1:0.2:180 (1 VAMP2, 'vNLP1'); 1:1:180 (four copies, 'vNLP4'); 1:2:180 (eight copies, 'vNLP8'); 1:4:180 (15 copies, 'vNLP15') and 1:8:180 (30 copies, 'vNLP30'). NLPs containing 1, 4 and 15 copies of VAMP-C45 were obtained using a similar approach. Excess detergent was removed using SM-2 bio-beads (Bio-Rad) overnight at 4°C with constant mixing. The assembled v-NLPs were separated from free proteins and lipids via gel filtration on a Superose six column (*Figure 1a*). Samples were concentrated using Amicon Ultra (50 KDa cutoff) centrifugal filter units, and analyzed by SDS-PAGE with Coomassie staining (*Figure 1b*). The number of VAMP2 copies per disc was determined by the VAMP2-to-ApoE ratio by densitometry using ImageJ (NIH). The number of ApoE copies per disc was estimated (*Bello et al., 2016*) using the calibration of disc size vs the number of ApoE copies previously reported (*Blanchette et al., 2008*). The size distribution of the v-NLPs was determined for every batch of production using transmission electron microscopy. To do this, the NLP discs were diluted (1:50), mounted onto carbon-coated 400 mesh copper electron microscopy grids, negatively stained with 2% uranyl acetate (w/v) solution, and subsequently examined in an FEI Tecnai-12 electron microscope operated at 120 kV. Micrographs of the specimen were taken on a Gatan Ultrascan4000 CCD camera at a magnification of 42,000. Typical micrographs and a size distribution are shown in *Figure 1c,d*. The size of the NLP discs with 1:180 ApoE422K: lipid ratio was typically 24 ± 2 nm (100–200 NLP discs were analysed for every production batch). Representative size distributions are shown as box plots for the conditions tested in *Figure 1e*. At least three independent batches of NLPs were used per condition. tNLPs were produced in a similar fashion, using a t-SNARE:ApoE:lipid ratio of 0.8:1:180 and 3:1:180 for tNLP4 (four copies of t-SNARE complex Stx/SN25 total per NLP) and tNLP 15 (15 total copies of t-SNAREs per NLP) samples, respectively.

## Bulk fusion of NLPs with t-SNARE liposomes

We used a previously established assay (*Bello et al., 2016*; *Shi et al., 2012*) to monitor the release of calcium from t-SNARE-reconstituted small unilamellar vesicles (t-SUVs) as they fused with discs

loaded with v-SNAREs. 40 µl t-SUVs entrapping 50 mM calcium were mixed with 5 µl vMSP NDs (prepared as described previously [*Shi et al., 2012*]) or with 10 µl of vNLP-discs in a buffer containing 2 µM of the calcium-sensitive dye mag-fluo-4 (Invitrogen, Carlsbad, CA). The mixture was loaded into a 96-well plate, and the mag-fluo-4 fluorescence ($\lambda_{ex}$= 480 nm, $\lambda_{em}$=520 nm, 515 cutoff) was recorded by a SpectraMax M5 plate reader (Molecular Devices, Sunnyvale, CA). After 60 min, 15 µl of 5% dodecylmaltoside was added and the mixture was incubated for an additional 20 min to release all remaining entrapped calcium and thus to establish the maximum mag-fluo4 signal. Fusion is reported as percent of maximum fluorescence in *Figure 1—figure supplement 1*.

## Single-cell lipid mixing and calcium-influx assays

These assays were carried out as described in *Wu et al. (2016)*.Briefly, for lipid mixing, tCells were plated in 35 mm poly-D-lysine-coated glass bottom dishes (MatTek Corporation, MA, USA) and vNLP8s were prepared as described above, except that one mole % each of 1,1'-dioctadecyl-3,3,3',3'-tetramethylindocarbocyanine perchlorate (DiI, cat. no. D282, Molecular Probes, Eugene, OR) and 1,1'-dioctadecyl-3,3,3',3'-tetramethylindodicarbocyanine perchlorate (DiD, cat. no. D307, Molecular Probes) fluorescent lipid labels were included in the lipid composition. For each reaction, 15 µl of vNLP8 was added onto tCells (final NLP lipid concentration was ~54 µM) and incubated for 30 min at 4°C, a temperature at which SNARE complexes assemble but cannot drive fusion (*Weber et al., 1998*). Fusion was started by raising the temperature to 37°C and monitored by the dequenching of the DiI fluorescence using confocal microscopy. As controls, empty nanodiscs (eNLP) or NLPs bearing VAMP2-4X (harboring the mutations L70D, A74R, A81D and L84D) were used (*Figure 2b*). To estimate the extent of lipid mixing, at the end of some experiments, DiD fluorescence was completely bleached using direct excitation at 647 nm with 100% laser power. This resulted in the maximum possible donor (DiI) intensity, $F_{max}$. We then rescaled the donor fluorescence values $F(t)$ to obtain the fraction of maximum DiI fluorescence: $\tilde{F}(t) = (F - F_0)/(F_{max} - F_0)$, where $F_0$ is the minimum at the beginning of acquision.

To assess lipid mixing, we also used an alternative protocol that avoided the cold incubation step; this protocol could not be used for time-course measurements because of the high background resulting from excess NLPs. The same amount of vNLP8 as above was added to tCells at 37°C. After 15 min incubation, excess NLPs were washed, and DiI, DiD and CFP fluorescence levels were acquired using confocal microscopy (*Figure 2c,d*).

To measure the influx of calcium through fusion pores, tCells were loaded with 5 µM of Fluo-4 AM (Life Technologies, Carlsbad, CA), a cell-permeant calcium-sensitive fluorescent dye, as previously described (*Wu et al., 2016*). After washing to remove dye that was not taken in by cells, vNLP (15 µl) were added to tCells at 37°C and the influx of calcium was tracked by imaging of Fluo-4 fluorescence using a confocal microscope (*Wu et al., 2016*).

## Electrophysiology

Details are given in *Wu et al. (2016)*. Briefly, flipped t-SNARE HeLa cells (tCells) were cultured in 3 cm dishes. For recordings, a dish was placed in a temperature-controlled holder (TC-202A by Harvard Apparatus (Holliston, MA), or Thermo Plate by Tokai Hit (Shizuoka-ken, Japan)) set at 37°C. Cells were visualized with an inverted microscope (Olympus IX71, Olympus Corp., Waltham, MA) using an Andor DU-885K EMCCD camera controlled by Solis software (Andor, South Windsor, CT). Recording pipettes (borosilicate glass, BF 150-86-10, Sutter Instruments, Novato, CA) were pulled using a model P-1000 pipette puller (Sutter Instruments) and polished using a micro-forge (MF-830, Narishige, Tokyo, Japan). Pipette resistances were 5–10 MΩ in NaCl-based solution. The bathing medium contained: 125 mM NaCl, 4 mM KCl, 2 mM CaCl$_2$, 1 mM MgCl$_2$, and 10 mM HEPES, (pH adjusted to 7.2 using NaOH) for the cell-attached recordings. 10 mM glucose was added to the medium before use. All voltage- and current-clamp recordings were made using a HEKA EPC10 Double USB amplifier (HEKA Elektronik Dr. Schulze GmbH, Lambrecht/Pfalz, Germany), controlled by Patchmaster software (HEKA). Currents were digitized at 20 kHz and filtered at 3 kHz.

To measure SNARE-mediated single fusion pore currents in the cell-attached mode (*Yang and Sigworth, 1998*), electrodes were filled with the pipette solution composed of 125 mM NaCl, 4 mM KCl, 10mM HEPES, 13 mM or 26 mM tetraethylammonium-Cl (TEA-Cl, K$^+$-channel antagonist), adjusted to pH 7.2 using NaOH. This solution had resistivity of 0.60 Ohm.m, measured using a

conductivity cell (DuraProbe, Orion Versa Star, Thermo Scientific). For experiments designed to test the presence of multiple pores connecting large copy number vNLPs to tCells (*Figure 5—figure supplement 4*), electrodes were filled with a solution containing: 129 mM N-methyl-d-glucamine (NMDG), 10 mM HEPES, 26 mM TEA-Cl, pH adjusted to 7.2 using HCl, resistivity 0.88 Ohm.m, 305 mOsm. The pipette tip was initially filled with 1 µl of NLP-free buffer and back-filled with vNLPs suspended in the same buffer (final [vNLP] = 100 nM, 120 µM lipids). This allowed the establishment of a tight seal ($R_{seal}$>10 GOhm) with high success rate, as well as the recording of a stable baseline before the vNLPs diffused to the membrane patch and started fusing with it 2–18 min later. Such a back-filling strategy is typically used in perforated patch measurements (*Sakmann and Neher, 2009*). All cell-attached recordings were performed using a holding potential of −40 mV relative to bath. With a cell resting membrane potential of $−56 \pm 7$ mV (mean ± S.D., n = 36), this provided 16 mV driving force across the patch membrane.

## Analysis of fusion pore data

The analysis of fusion pores is described in detail in *Wu et al. (2016)*. Briefly, we developed an interactive graphical user interface in Matlab to help to identify, crop and process single fusion pore currents. Traces were exported from Patchmaster (HEKA Electronik) to Matlab (Mathworks) and low-pass filtered (280 Hz cutoff); frequencies that were the result of line voltage were removed using notch filtering. Zero phase shift digital filtering algorithms (Matlab Signal Processing Toolbox function filtfilt) were employed to prevent signal distortion. Filtered traces were averaged in blocks of 80 points (125 Hz final bandwidth) to achieve rms baseline noise $\leq 0.2$ pA. Currents $I$ for which $|I|>2.0$ pA for at least 250 ms were accepted as fusion pore current bursts. During a burst, rapidly fluctuating currents often returned to baseline multiple times, i.e. pores flickered. To quantify pore flickering, we defined currents $< -0.25$ pA and lasting $\geq 60$ ms (15 points) as open pores and currents not meeting these criteria as closed. For a given burst, the number of open periods was equal to the number of flickers, $N_{flickers}$. To estimate the fusion rate for each recording (i.e. the rate at which current bursts appeared), we counted the number of current bursts that fit the set criteria (current amplitude >2 pA for at least 250 ms) and divided this number by the duration of the recording. Rates from different records (patches) were averaged for each condition. We also refer to this rate as the pore nucleation rate. Periods during which the baseline was not stable were excluded from this analysis. Many recordings ended with what seemed to be currents from overlapping fusion pores. Such end-of-record currents were also excluded because they could also be attributed to a loose seal. Thus, the fusion rates that we report may underestimate the true rates, especially for conditions in which fusion activity was high. For distributions of conductances and radii, we used pore open-state values, denoted by the subscript 'po'. For *Figure 4e,f*, we first computed the probability density functions (PDFs) for individual pores using a fixed bin width for all, then averaged these to give equal weight to all pores. All distribution fitting was performed using Matlab Statistics Toolbox functions fitdist or mle, using maximum likelihood estimation. Open-pore conductance values were used point-by-point to estimate the open-pore radii, by approximating the pore as a cylinder and using the expression (*Hille, 2001*) $r_{po} = \left( \rho \lambda G_{po}/\pi \right)^{1/2}$, where $\rho$ is the resistivity of the solution, $\lambda = 15$ nm is the length of the cylinder, and $G_{po}$ is the open-pore conductance. For assessing statistical significance when comparing sample means, we used the two-sample t-test or the nonparametric two-sample Kolmogorov-Smirnov test (ttest2 or kstest2, Matlab Statistics Toolbox), as indicated in the figure legends. We considered each single-pore measurement to be a biological replicate.

For clustering average single-pore conductances $\langle \mathrm{G}_{po} \rangle$ for vNLP30 measurements, we used a two-component Gaussian mixture model (*Figure 6b*) that indicated a boundary between the two components at ~1 nS. Applying this cutoff to all vNLP samples, we produced the boxplot in *Figure 6c*, where the central red line on each box marks the median, the edges of the box are the 25th and 75th percentiles, and the whiskers extend from q3 +1.5(q3 – q1) to q1 – 1.5(q3 – q1), where q1 and q3 are the 25th and 75th percentiles, respectively. For a given v-SNARE copy number per NLP face, $N_{SNARE}$, we defined the probability $P_{dilation}$ of achieving a high-conductance pore ($\langle G_{po} \rangle$>1 nS) as the fraction of high conductance pores observed for that copy number. For example, only 3 out of 64 pores were large conductance for vNLP8, which had four copies per face (*Figure 6c*), hence $P_{dilation}(N_{SNARE} = 4) = 3/64$. We plotted $P_{dilation}$ as a function of v-SNARE copy number per NLP face in *Figure 6d*. In *Figure 7a*, to estimate the energy profiles of fusion pores for

a given v-SNARE copy number, we first calculated the probability density function for open-pore radii as in *Figure 4f*. The probability $P_r$ that the radius is between $r$ and $r + \Delta r$ is the density at that bin $\times \Delta r$, where $\Delta r$ is the bin width. We estimated the energy $U(r)$ of a pore with radius $r$ by $U/kT = -ln(P_r) + A$, where $A$ is an arbitrary constant.

## Mathematical model of the fusion pore between a nanodisc and planar membrane in the presence of SNAREs

### Membrane free energy

We modeled the fusion pore as having a toroidal shape formed between a nanolipoprotein particle (NLP) modeled as a planar bilayer of diameter $D$ and the tCell membrane modeled as an infinite planar bilayer, both of which are at a constant membrane tension (*Figure 7c*). This toroidal assumption is similar to that in previous theoretical studies that assumed a toroidal shape of the fusion pore (*Chizmadzhev et al., 2000*; *Jackson, 2010*; *Kozlov et al., 1989*). Experimental studies also observed an hourglass-shaped fusion pore that could be considered approximately toroidal (*Curran et al., 1993*; *Haluska et al., 2006*).

The fusion pore is parametrized by the toroidal shape parameters: the radius of the toroid $r_{po}$, which corresponds to the fusion pore radius $r_{po}$, and the separation of the membranes at the edge of the NLP $h$. The fusion pore is completely toroidal at small pore sizes. The free energy of the fusion pore is calculated using the Helfrich energy form, as was used in previous studies (*Chizmadzhev et al., 2000*; *Jackson, 2010*; *Kozlov et al., 1989*)

$$U_{mb}(r_{po}, h) = U_{bend}(r_{po}, h) + \gamma \Delta A(r_{po}, h) \tag{1}$$

Here, the energy due to bending is given by

$$U_{bend}(r_{po}, h) = \frac{\kappa}{2} \int_M (2C)^2 dA \tag{2}$$

where $\kappa$ and $C$ are the bending modulus and the mean curvature of the membrane, respectively. The energy expended to add membrane area due to membrane tension $\gamma$ is the second term, where $\Delta A$ is the change in total membrane area due to pore formation, given by $\Delta A(r_{po}, h) = A_{po}(r_{po}, h) - A_{rim}(r_{po}, h)$. Here, $A_{po}$ is the area of the fusion pore. $A_{rim}$ is the area of both rims of the fusion pore, which is the area that has to be removed from the infinite tCell membrane and the NLP membrane to form the pore. We evaluated all integrals and all areas over the midplane $M$ of the membrane forming the pore to give:

$$U_{bend}(r_{po}, h) = \pi\kappa \left\{ \frac{2(R+H)^2}{H\sqrt{R(R+2H)}} \tan^{-1}\left(\sqrt{\frac{R+2H}{R}}\right) - 4 \right\} \tag{3}$$

$$\Delta A(r_{po}, h) = \pi H((2\pi - 4)H + 2\pi R) - 2\pi(H+R)^2 \tag{4}$$

where $H = h/2 + \delta/2$ and $R = r_{po} + \delta/2$. Values of $\kappa$ in previous studies range from 10–50 $kT$ (*Cohen and Melikyan, 2004*; *Brochard and Lennon, 1975*; *Khelashvili et al., 2013*; *Marsh, 2006*). A commonly used value is $\kappa = 20kT$ (*Jackson, 2009*), which we used here (*Table 1*). $\gamma$ was obtained as a best-fit parameter (*Table 1*).

### Free energy due to twisting of the ApoE proteins that line the boundary of the ND

Owing to the finite size of the NLP, toroidal states are not possible for large pores with $r_{po}$ and $h$ that constitute a sizeable fraction of the NLP diameter, $D$. These shapes are partially toroidal and come into existence when $r_{po} + h/2 + \delta \geq D/2$, where $\delta$ is the membrane thickness. The ApoE proteins that line the NLP boundary need to be rotated through an angle $\phi$ to form these shapes. An example of one such shape is the right-hand side of *Figure 7c*.

We assumed these proteins exert a constant torque $\tau$ per unit length of the NLP boundary to resist this rotation. The ApoE proteins exert no torque in the completely toroidal states as $\phi$ vanishes for these states. The free energy of these proteins is

$$U_{\mathrm{ApoE}}\left(r_{\mathrm{po}}, h\right) = \tau \pi D \phi \tag{5}$$

$$\phi\left(r_{\mathrm{po}}, h\right) = \sin^{-1}\left(\frac{r_{\mathrm{po}} + \frac{h}{2} + \delta - \frac{D}{2}}{\frac{h}{2}}\right), \ r_{\mathrm{po}} + \frac{h}{2} + \delta \geq \frac{D}{2} \tag{6}$$

Thus, the ApoE proteins resist pore expansion as $\phi$ increases with the size of the pore. We obtained $\tau$ as a fitting parameter and $D$ was measured in this study (values in *Table 1*). For further details about the partially toroidal states, please see the penultimate subheading '*Description of partially toroidal states and calculation of membrane free energy*'.

## Free energy contribution from short-ranged steric-hydration forces

Steric-hydration repulsion between membranes is prominent at small membrane separation. Experimentally measured steric-hydration pressures between planar membranes of separation $d$ are of the form $P_0 \exp(-d/\lambda)$ (*Rand and Parsegian, 1989*). Values of $P_0$ and $\lambda$ have been measured before for several membrane compositions; $\lambda$ is within 0.1–0.3 nm (*Rand and Parsegian, 1989*). As the pore sizes over which these effects are appreciable ($\sim\lambda$) are very small compared with the NLP diameter $D$, only toroidal states are considered for this calculation. We obtained $P_0$ from previous studies and $\lambda$ as a best-fit parameter (*Table 1*).

The steric-hydration forces act in two orthogonal directions on the membranes comprising the fusion pore: to increase the pore radius ($r_{\mathrm{po}}$) and the separation between membranes ($h$). The sum of the work done by these two forces gives the free energy of the steric-hydration interaction

$$U_{\mathrm{hyd}}\left(r_{\mathrm{po}}, h\right) = P_0 \lambda \left(\frac{\pi D^2}{4}\right) \exp\left(-\frac{h}{\lambda}\right) + P_0 (2\pi l) \exp\left(-\frac{2r_{\mathrm{po}}}{\lambda}\right) \left(\frac{\lambda}{2} r_{\mathrm{po}} + \left(\frac{\lambda}{2}\right)^2\right) \tag{7}$$

where $l$ is the effective pore height, i.e. the height of the section of the pore that contributes substantially to the steric-hydration interaction. To obtain the free energy contribution of the steric-hydration forces, we now calculate the work done by these forces to assemble the fusion pore. We first observe that the work done ($W$) to bring two patches of membranes of area $\delta A$ to a separation $h$ from a large distance apart is

$$W = P_0 \int_{\infty}^{h} \delta A \exp\left(-\frac{y}{\lambda}\right) dy \tag{8}$$

The first term in *Equation 7* is the work done to separate the planar part of the membranes to a distance $h$. As the pore area where these forces are relevant ($\sim\lambda^2$) is very small compared with the NLP area $\pi D^2/4$, we set the area of the planar region $\delta A = \pi D^2/4$ in *Equation 8* to obtain the first term. The second term in *Equation 7* is the work done to separate the membranes to form a pore of diameter $2r_{\mathrm{po}}$. To calculate this, we can imagine that the pore is a cylinder of diameter $2r_{\mathrm{po}}$ and height $l$, where $l = \sqrt{2\lambda(h + 2\delta)}$, since the change in pore diameter over the height $l$ is negligible. Here, $\delta$ is the thickness of the bilayer and $l$ is the height over which the cross-sectional diameter of the toroidal shape increases from $2r_{\mathrm{po}}$ to $2r_{\mathrm{po}} + \lambda$.

To obtain $l$, we consider the inner surface of the toroidal pore. This shape is formed by revolving the semicircle given by $x = (r_{\mathrm{po}} + R') - R' \cos\theta, z = R' \sin\theta$ where $-90^0 \leq \theta \leq 90^0$ in the XZ plane about the Z axis. Here, $r_{\mathrm{po}}$ is the radius of the pore and $R'$ is the radius of the semicircle, which is also equal to half of the maximum separation between the heads of the monolayers that line the inner surface of the pore, $R' = h/2 + \delta$ as can be seen from *Figure 7c*. $l/2$ is that value of $z$ at which the cross-sectional radius of the pore $x$ increases to $r_{\mathrm{po}} + \lambda/2$. Thus, $l$ is obtained by solving the equation of the semicircle $\left(r_{\mathrm{po}} + h/2 + \delta - r\right)^2 + (l/2)^2 = (h/2 + \delta)^2$ to first order in $\lambda$ where $r = r_{\mathrm{po}} + \lambda/2$.

To calculate the steric-hydration contribution from a pore of size $r_{\mathrm{po}}$ and height $h$, one need only consider the pore over that height $l$ at which the cross-sectional diameter of the toroidal shape increases from $2r_{\mathrm{po}}$ to $2r_{\mathrm{po}} + \lambda$, since $\lambda$ is the range of the steric-hydration force. Thus, the area of

the cylinder is $\delta A(r_{po}) = 2\pi r_{po}l$, and using **Equation 8,** the work done to set up the pore is $P_0 \int_{\infty}^{2r_{po}} (2\pi ly)\exp(-y/\lambda)dy$, giving the second term of **Equation 7**.

## Free energy contribution from SNARE proteins

In this section, we calculate the free energy due to the SNARE proteins. We fix the total number of v-SNAREs. Out of these $N$ v-SNAREs, $N_z$ are fully zippered and $N_u$ are partially zippered. Only the TMDs and the linker regions of these partially zippered SNAREs are unzipped. For each $N$, we allowed $N_z$ and $N_u$ to vary from 0 to $N$ to obtain an equilibrium distribution for both. We calculated the free energy of the SNAREs for $N = 0, 2, 4, 8, 15$, which correspond to the total number of v-SNAREs per NLP face used experimentally (**Figure 7a**). To match with experiment, we used the assumption that only half of the total number of SNAREs present in the NLP would be present on the side of the NLP that faces the tCell, and that all of these SNAREs would be available to participate in fusion pore expansion.

We assumed that the fully zippered SNAREs form a ring at the waist of the fusion pore (**Figure 7c**). Their free energy is

$$U_z\left(r_{po}, N_z\right) = -N_z kT \left(\ln\frac{2\pi r_{po} - N_z b}{b} + 1\right) - N_z\varepsilon_{zip} - N_z kT \ln\Omega_z \tag{9}$$

The first term is the positional entropy of the zippered TMDs whose diameter is $b$, which we measured by fitting a cylinder to the measured crystal structure (**Stein et al., 2009**). The second term is the energy released when a partially zippered SNAREpin completes its zippering. The third term is the orientational entropy associated with the zippered SNAREs. We assume that these are very stiff cylindrical rods. Due to their high stiffness, these rods can only explore a small solid angle $\Omega_z = 0.05\text{sr}$ (**Table 1**). We calculated this angle based on an atomistic molecular dynamics study of the t-SNARE TMD that shows that these domains explore angles of ~$10^0$ around their equilibrium position in a bilayer (**Knecht and Grubmüller, 2003**). The SNAREs fluctuate about their equilibrium orientation, which we assume is the local normal to the membrane.

The free energy of the partially unzipped SNAREs is

$$U_{uz}(N_{uz}) = -N_{uz}kT\left(\ln\frac{2\pi D}{b}\right) - N_{uz}kT\ln\pi \tag{10}$$

The first term is the positional entropy of the TMDs. These partially zippered SNAREs are in a Y-shape with both unzipped TMD domains on the same side of the pore, either on the vNLP or the tCell membrane. The linker domains are also unzipped and this imparts flexibility to these SNAREs (**Jahn and Scheller, 2006**). This orientational freedom is given by the second term. These SNAREs can adopt all orientations in which they do not intersect with the membrane; this corresponds to a solid angle of $\pi$ steradian. We restricted these SNAREs to a circle of radius equal to that of the NLP, as this considerable orientational freedom is only available when the SNAREpin body is away from the fusion pore lumen.

## Calculation of the total free energy $U$ as a function of pore size and number of SNAREs

The probability that a fusion pore accesses a radius $r_{po}$ in the presence of $N$ SNAREs is proportional to $\exp(-U(r_{po}, N)/kT)$ in equilibrium, as this is the Boltzmann distribution where $U$ is the total free energy. We assumed that the bilayer-SNARE system is equilibrated as the current bursts measured experimentally are long-lived. To calculate this free energy, we summed the Boltzmann factor of all states that comprise such a system:

$$\exp\left(-\frac{U\left(r_{po}, N\right)}{kT}\right) = \sum_{N_z=0}^{N} \int_{b}^{\infty} \exp\left(-\frac{U_{tot}\left(r_{po}, h, N, N_z\right)}{kT}\right)dh \tag{11}$$

Here, $U_{tot}$ is the total free energy of one fusion pore state with $N_z$ zippered SNAREs and of membrane separation $h$, given by

$$U_{\text{tot}} = U_{\text{mb}} + U_z + U_{\text{uz}} + U_{\text{hyd}} + U_{\text{ApoE}} \tag{12}$$

We performed the integration and the sum over all states in *Equation 11* numerically in MATLAB.

## Derivation of best-fit model parameters by fitting model-predicted free energy to experiment

We performed a numerical calculation using *Equation 11* to obtain free -energy curves as a function of SNARE copy numbers and pore size. For the membrane parameters, it is best to fit to data from membranes with no SNAREs. We fit the fusion pore free energy predicted by the model with no SNAREs (setting $N = 0$ in *Equation 11*) to the experimentally measured curve for one SNARE, assuming that such a pore behaves similarly to a protein-free pore. We first fit the model-predicted pore size at the minimum in the free energy to experiment by using the steric-hydration force scale $\lambda$ as a best-fit parameter. We obtained the bending modulus $\kappa$ from *Jackson (2009)* and fit the slope following the minimum in the region $0.5\text{nm} \leq r_{\text{po}} \leq 1.0\text{nm}$ using the membrane tension $\gamma$ as a best-fit parameter, as $\gamma$ largely determines the slope beyond this minimum.

Using these parameters, we calculated the free energy versus pore radius in the presence of SNAREpins (*Figure 7b*). As SNAREpins are introduced, the model predicts that the minimum barely shifts, while the slope beyond the minimum decreases with increasing numbers of SNAREs by an amount depending on the zippering energy parameter $\varepsilon_{\text{zip}}$. We selected a typical experimental curve (vNLP30) and fit the slope of the free energy from simulation to that measured from experiment for $0.5 \text{ nm} \leq r_{\text{po}} \leq 2.5\text{nm}$ and $N = 15$ SNAREs (as vNLP30 corresponds to 15 SNAREs per face) using $\varepsilon_{\text{zip}}$ as a fitting parameter, and obtained $\varepsilon_{\text{zip}} = 9.6\text{kT}$. This is higher than the zippering energy of the linker domains alone, which was measured to be ~5 kT by *Gao et al. (2012)*.

Further increases in pore size cause increases in free energy, as the fusion pore shapes are partially toroidal and the twisting torque from the ApoE proteins at the NLP boundary resists further expansion. Thus, we fit the slope of the free energy curve in this region ($4\text{nm} \leq r_{\text{po}} \leq 4.5\text{nm}$) measured from simulation to that measured experimentally for $N = 4$ SNAREs to obtain the torque per unit length $\tau = 8.43$ pN as a fitting parameter.

To understand how lipid-anchored VAMP2 affects pore dilation, we reduced the size of the zippered TMDs by 50% as the zippered SNARE complex lacks the v-SNARE TMD. This is the maximum possible reduction in the excluded volume of the zippered SNAREs. We then varied the zippering energy to ensure that the model-predicted mean pore size was invariant with respect to this reduction, consistent with the invariance in mean pore size observed in experiments (*Figure 5—figure supplement 2f*). The best-fit zippering energy is 0.43 kT lower than the best-fit value obtained when both TMDs are present. This is an upper bound on the reduction in zippering energy given that we used the largest possible reduction in the excluded volume.

## Description of partially toroidal states and calculation of membrane free energy

Due to the finite size of the NLP, toroidal states are not possible for large pores. We instead assumed that these shapes are partially toroidal. These come into existence when $r_{\text{po}} + h/2 + \delta \geq D/2$, where $\delta$ is the membrane thickness. We set the shape of these states as follows. We constructed a toroidal pore with the shape parameters $r_{\text{po}}, h$, and truncated the top half of the toroid at the plane where the cross-sectional diameter of the toroid is equal to the NLP diameter (*Figure 7c*, right panel). In these partially toroidal states, the ApoE proteins at the edge of the NLP are rotated through an angle $\phi$ compared with the fully toroidal states (*Equation 6*).

Evaluating the integral from *Equation 2*

$$U_{\text{bend}}(r_{\text{po}}, h, \phi) = U_{\text{bend}}(r_{\text{po}}, h, 0) + 2\pi\kappa(1 - \cos\phi) - \pi\kappa\frac{(R+H)^2\tan^{-1}(\alpha)}{H\sqrt{R(R+2H)}} \tag{13}$$

where $\quad H = h/2 + \delta/2, R = r_{\text{po}} + \delta/2, \tan^{-1}\alpha = \tan^{-1}\left(\sqrt{\frac{R}{R+2H}}\cot\left(\frac{\pi-2\phi}{4}\right)\right) - \tan^{-1}\left(\sqrt{\frac{R}{R+2H}}\right),\quad$ and

$U_{\text{bend}}(r_{\text{po}}, h, 0)$ is the bending energy in a fully toroidal state, given by **Equation 3**. The change in total membrane area due to pore formation in these partially toroidal states is given by

$$\Delta A_{\text{po}}(r_{\text{po}}, R_{\text{po}}, \phi) = \pi H((R+H)(2\pi-\phi) - 2H(1+\cos\phi)) + \pi(R+H)^2 - \pi\left(\frac{D}{2}\right)^2 \tag{14}$$

In the partially toroidal states, $h$ does not correspond to the membrane separation at the edge of the pore, although $r_{\text{po}}$ is still the pore radius. Thus, we indicate the membrane separation at the NLP boundary by $h_{\text{po}}$ given by

$$h_{\text{po}} = \begin{cases} \frac{h}{2}(1+\cos\phi), & r_{\text{po}} + \frac{h}{2} + \delta \geq \frac{D}{2} \\ h, & r_{\text{po}} + \frac{h}{2} + \delta < \frac{D}{2} \end{cases} \tag{15}$$

## Acknowledgements

We thank F Sigworth and D Zenisek for expert advice and discussions, and J E Rothman, M Caplan, and members of the Karatekin and Rothman laboratories for helpful discussions. We are grateful to James E Rothman, Frederick Sigworth, David Zenisek, Yongli Zhang, and Thomas Melia for critically reading the manuscript. Jing Wang kindly provided C45 lipids. This work was supported by NIH grant R01GM108954 and a Kavli Neuroscience Scholar Award to EK. SMA, OB, and SK are members of the Rothman laboratory, supported by NIH R01DK027044. WV was supported by the Deutsche Forschungsgemeinschaft (DFG, German Research Foundation), fellowship VE760/1-1.

## Additional information

### Funding

| Funder | Grant reference number | Author |
|---|---|---|
| National Institute of General Medical Sciences | R01GM108954 | Erdem Karatekin |
| Kavli Foundation | Neuroscience Scholar Award | Erdem Karatekin |
| Deutsche Forschungsgemeinschaft | VE760/1-1 | Wensi Vennekate |
| National Institute of Diabetes and Digestive and Kidney Diseases | R01DK027044 | Shyam S Krishnakumar |

The funders had no role in study design, data collection and interpretation, or the decision to submit the work for publication.

### Author contributions

ZW, Conceptualization, Data curation, Formal analysis, Investigation, Visualization, Methodology, Writing—review and editing; ODB, Data curation, Formal analysis, Investigation, Methodology, Writing—review and editing; ST, Software, Formal analysis, Investigation, Visualization, Methodology, Writing—review and editing; SMA, Data curation, Formal analysis, Investigation, Visualization, Writing—review and editing; WV, Formal analysis, Writing—review and editing; SSK, Supervision, Investigation, Methodology, Writing—review and editing; BO, Software, Formal analysis, Supervision, Validation, Investigation, Methodology, Writing—review and editing; EK, Conceptualization, Data curation, Software, Formal analysis, Supervision, Funding acquisition, Investigation, Visualization, Methodology, Writing—original draft, Project administration, Writing—review and editing

### Author ORCIDs

Sathish Thiyagarajan, http://orcid.org/0000-0001-8545-8886
Shyam S Krishnakumar, http://orcid.org/0000-0001-6148-3251
Erdem Karatekin, http://orcid.org/0000-0002-5934-8728

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
