## [Decision Letter]

Thank you for submitting your article "Dilation of fusion pores by crowding of SNARE proteins" for consideration by *eLife*. Your article has been favorably evaluated by Randy Schekman (Senior Editor) and three reviewers, one of whom is a member of our Board of Reviewing Editors. The following individuals involved in review of your submission have agreed to reveal their identity: Josep Rizo (Reviewer #2); Jiajie Diao (Reviewer #3).

The reviewers have discussed the reviews with one another and the Reviewing Editor concluded that major revisions are required before a final decision can be made.

Summary:

This study shows how the number of SNARE complexes affects the formation of fusion pores between discoidal lipid nanoparticles containing VAMP2 (vNPLs) and cells expressing flipped t-SNAREs at the plasma membrane. The approach used in this work is an extension of previous work (Wu et al., 2016) but now with larger (21-27 nm diameter) nanolipoprotein particles (nlp) instead of the smaller nanodiscs used in the previous work (6-18 nm diameter). The larger nlp discs can accommodate more v-SNARE proteins. The electrophysiological approach used by the authors offers better time resolution than optical microscopy approaches used to date to study fusion in reconstituted systems. Their data show how pores form and flicker back and forth for long periods of time. Even relatively large numbers of SNAREs yield pores of limited conductance. However, there are a number of potential concerns that limit the insights into SNARE mediated fusion. Overall, this appears to be more of a methods paper and the conclusions regarding SNARE mediated fusion should be toned down.

Essential revisions:

1) From the example shown in Figure 5, it appears all fusion pores eventually reseal. Why is that? Could this be a consequence of cellular resealing mechanisms or constraints imposed by the patch clamp? Please test this possibility by using different size clamps.

2) These experiments use flipped t-SNAREs which exhibit considerably slower kinetics than wild type SNAREs (Giraudo et al., 2006): fusion in the presence of synaptotagmin and Ca^2+^ occurs in several minutes, vs. msec to sec. in reconstituted systems with wildtype SNAREs and synaptotagmin. Thus, use of the flipped t-SNAREs may profoundly alter the kinetics of the fusion pore opening and dilation as well as the number of SNARE complexes required to promote fusion. Moreover, the lipid composition of the outer leaflet of the cell membrane may be quite different from that of the inner leaflet of the plasma membrane (there is little PS in the outer leaflet of the cell membrane). Ideally, the authors should consider experiments to alleviate these concerns, but the minimum, these limitations need to be discussed in detail.

3) Another limitation of this study is that it is focuses on SNAREs only. Prior experiments have been done with a combination of large and small fluorescent probes to measure fusion pore dilation, e.g.: (Lai et al., 2013), and concluded that factors such as synaptotagmin are required for efficient pore expansion. Moreover, with the help of these other proteins, the number of SNARE complexes needed for fusion could be much smaller. The authors do mention accessory proteins at the end of the manuscript, but only in the context of organizing SNARE complexes, as indicated by the term 'accessory' itself, and without considering a direct role in membrane fusion. At the minimum, the authors should tone down every conclusion regarding the number of SNAREs required for neurotransmitter release.

4) Please indicate the percentage of vNLPs that fuse with the cells for the different vNLPs. This is a critical issue when rationalizing the data in terms of how the number of SNAREs influence fusion pore properties because the vNPLs are expected to have a distribution of VAMP copies rather than a single number. If the percentage of vNLPs that is low, the fusion may arise from the population that has much higher VAMP copies than assumed from the average. Describing the percentage of vNPLs that fuse is also important to evaluate this overall approach.

5) The number of fusion pores/min observed with empty NLPs (is what eNLP means?) is low compared to that observed with vNLP8, but not negligible; it appears to be about 8 times smaller in Figure 4. Does this mean that eNLPs fuse spontaneously with the cells? If this is the case, an acceleration by a factor of 8 suggests that 8 SNARE complexes only provide about 1 kcal/mol to facilitate membrane fusion. If there is no flaw in this argument, the authors should emphasize this point and compare this energetic estimate will all the energetic arguments they make later in the manuscript.

6) Conclusions regarding fusion pore expansion should be taken with caution because the scaffolding protein may impose constraints that would not be present in vivo.

7) A concern is about the possibility of formation of multiple pores. In the third paragraph of the Results, the authors provided an explanation for single pore formation. However, they apparently excluded the possibility of pore formation involving multiple v-SNARE nanolipoprotein particles at the same time. Due to the large size of nanolipoprotein particles (23 nm in diameter), it is possible that there is more than one fusion pore for individual nanolipoprotein particle in the presence of many SNAREs. For example, compared to the large jump from 15 to 30 v-SNAREs, the difference between 8 and 15 v-SNAREs is negligible (Figure 5). Could this imply the formation of another fusion pore?

8) In the section 'Derivation of best-fit model parameters…', the authors showed how they obtain the parameters in the model by fitting the calculated results of the model (equation 11) to the experimental data. For example, they fit the results in the range of 0.2 nm ≤ rpo ≤ 1.5 nm to obtain the tension of membrane γ and the steric-hydration force length scale λ (0.5 nm ≤ rpo ≤ 2.5nm) to determine a range for epsilon (4 nm ≤ rpo ≤ 4.5nm) to obtain tau. However, in equation 11, the total free energy depends on all the parameters in the full range of rpo, so how can the authors obtain an individual parameter in a different range of rpo?

9) The parameters in the mathematical model (i.e., rpo, h, δ, D, and the twisting angle phi of ApoE proteins) should be explicitly shown in the model in Figure 7.

10) To calculate the second term in equation 7, the authors approximated the pore as a cylinder of radius rpo and height λ = 0.13 nm. It seems that the height of the cylinder is too small, as the neck of the fusion pore can be on the order of ~5 nm in height. Please explain why choosing such a small height for the cylinder is a reasonable approximation.

[Editors' note: further revisions were requested prior to acceptance, as described below.]

Thank you for resubmitting your work entitled "Dilation of fusion pores by crowding of SNARE proteins" for further consideration at *eLife*. Your revised article has been favorably evaluated by Randy Schekman (Senior editor) and three reviewers, one of whom is a member of our Board of Reviewing Editors.

The manuscript has been improved but there are two remaining issues that need to be addressed before acceptance, as outlined below, using the same bullet numbers used for the points of the decision letter.

2) Some of the arguments offered with regard to lipid composition are reasonable, but saying that 'short PI(4,5)P2 does not have an effect (data not shown)' is not convincing, and the problem still remains that it is difficult to control lipid composition in this system and make it similar to physiological. This is particularly important considering the low percentage of fusion that they now report (point 4). The authors cite Giraudo et al., 2006, but the work in that paper suffers from the same problem. This concern does not invalidate the results from the authors, but they should acknowledge the problem in the manuscript. The fact is that lipids could play key roles in the fusion mechanism and the tendency of SNARE-centric models of membrane fusion to ignore this fact is deleterious to the scientific discussion in the field.

4) The authors again make some good arguments, but they really do not address the heart of the problem. With such low percentage of fusion, it seems very likely that fusion occurs for low populations of nanodiscs with higher copy number of SNAREs than the mean. The authors claim that this is unlikely because 'the fusion rate does not increase for mean copy number greater than 2-4 per NLP side (Figure 5)'. However, in Figure 5 one can see a tendency for the fusion rate to increase up to vNLP15. Even though the differences may not be statistically significant, one just cannot draw the conclusion written by the authors. Hence, the authors should explicitly acknowledge that the actual numbers of SNAREs underlying fusion may be (in fact that are very likely to be) higher than the mean values described. The same issue applies to measurements of conductance.

---

## [Author Response]

*Essential revisions:*

*1) From the example shown in Figure 5, it appears all fusion pores eventually reseal. Why is that? Could this be a consequence of cellular resealing mechanisms or constraints imposed by the patch clamp? Please test this possibility by using different size clamps.*

The simplest explanation for pore closure is that the pore does not have an alternative. It cannot expand beyond a maximum size due to constraints imposed by the scaffold protein. It also does not remain open indefinitely, because presumably the pore is a higher energy structure than a resealed membrane (either a hemi-fission structure or two separate bilayers). This is consistent with the fact that it is difficult to capture fusion pores in various experiments (e.g. fusion pores were not detected in EM studies by Hernandez..Jahn Science, 2012, 336:1552, Diao…Brunger *eLife*, 2012, 1:e00109, or Shi…Pincet, Science 2012, 335:1355). So we expect the pore to reseal at some point given enough time, independent of any cellular response of contraints imposed by the cell-attached patch clamp method.

Indeed, previous work that studied fusion between v-SNARE nanodiscs to t-SNARE liposomes also found pores eventually reseal. Both the Rothman (Shi et al., Science, 2012; Bello et al.,) and the Chapman labs (Bao et al., 2016) showed that addition of dithionite after the fusion reaction quenched only a fraction of the NBD-PE lipid labels. Had the pores remained open, the quencher (0.2-0.3 nm Stokes radius) would have had access to the liposome lumen and quenched all the signal. In addition, both laboratories found at least some of the nanodiscs attached to the liposomes, perhaps in a hemi-fission state.

To exclude artifacts due to the cell-attached recording configuration, in previous work (Sci. Rep. 6:27287, 2016) we patched flipped t-SNARE cells in the whole cell configuration, puffed nanodiscs onto the cell using a pressure-driven perfusion system, and monitored whole-cell currents. We observed large currents upon perfusion of discs, consistent with disc-cell fusion.

The currents returned to baseline upon cessation of disc application, indicating pores resealed.

To test if pores also reseal in our calcium influx assay, we have performed new experiments in which we loaded cells with a calcium indicator, Fluo-4, then added nanodiscs. The Fluo-4 fluorescence increased, consistent with fusion allowing calcium entry into the cells, as in Figure 3. We then washed away unbound discs, and continued to monitor Fluo-4 signals which returned to the baseline after a few minutes. These results are consistent with our whole-cell recordings mentioned earlier and suggest that fusion pores connecting nanodiscs to cells do close, even in the absence of any patch-clamp recordings. We have added these new results as Figure 3—figure supplement 1.

Although it is clear that the pore must reseal, it is still possible that the pore lifetime is affected by the patch method or cellular processes. In our cell-attached recordings of single-pore currents, we do not expect cellular wound healing mechanisms to contribute significantly. Such repair mechanisms have been studied in response to much larger pores (~ 1 micron or larger). In addition, a large extracellular calcium concentration (~1 mM or larger) is an essential requirement for cell wound healing (Moe, Golding, and Bement, Sem. Cell. Dev. Biol., 2015, 45:1823), but our pipette solutions did not contain any calcium.

Finally, an indication that cellular processes do not have an appreciable effect on pore lifetimes is that the largest effect on pore lifetimes so far was obtained by three point mutations in the v- SNARE transmembrane domain (Wu et al., 2016), prolonging pore lifetimes 10-fold. It is hard to imagine that point mutations on the v-SNARE TMD would affect any cellular processes.

*2) These experiments use flipped t-SNAREs which exhibit considerably slower kinetics than wild type SNAREs (Giraudo et al., 2006): fusion in the presence of synaptotagmin and Ca^2+^ occurs in several minutes, vs. msec to sec. in reconstituted systems with wildtype SNAREs and synaptotagmin. Thus, use of the flipped t-SNAREs may profoundly alter the kinetics of the fusion pore opening and dilation as well as the number of SNARE complexes required to promote fusion. Moreover, the lipid composition of the outer leaflet of the cell membrane may be quite different from that of the inner leaflet of the plasma membrane (there is little PS in the outer leaflet of the cell membrane). Ideally, the authors should consider experiments to alleviate these concerns, but the minimum, these limitations need to be discussed in detail.*

The slow fusion kinetics in the cell-cell fusion experiments in Giraudo et al., which included flipped synaptotagmin and GPI-anchored complexin in addition to flipped SNAREs, was almost certainly limited by the detection method used. First, the classification of cells as fused required complete mixing of cytoplasms and the uniform appearance of dsRed (cytosolic), CFP (nuclear) and YFP (nuclear) signals, which typically requires minutes after fusion. Second, rather than performing live-cell imaging, the authors fixed the cells after a given incubation period, mounted them, and collected confocal images at a later time. This allowed several experiments and controls to be run in parallel in multiwell plates, at the expense of high-resolution kinetic measurements. The assay was designed to acquire images of large numbers of cells, using a small magnification objective (20x) for most experiments. This approach precluded analysis of the true kinetics of the fusion reaction, which was presumably much faster than the minutes time scale that could be probed in the assay.

In sharp contrast to cell-cell fusion experiments, much faster time scales can be probed in liposome-liposome fusion experiments, because the volumes and areas involved for content and lipid mixing are much smaller in the liposomes than in cells and diffusion is not hindered by cytosolic or membrane proteins.

It is true that the lipid composition of our target membrane (outer leaflet of the plasma membrane) is very different from that of the physiological target membrane (the inner leaflet of the plasma membrane). This is a potential issue we tested as follows. First, we flipped the configuration and fused flipped v-SNARE cells with t-SNARE NLPs. This allowed us to have a better mimic of the inner plasma membrane leaflet composition on the target membrane (now the tNLP membrane). This resulted in similar fusion rates and pore properties for two different SNARE copy numbers per NLP. This new data is now added as Figure 5—figure supplement 3 and mentioned in the main text.

Second, we added short-chain PI(4,5)P2 to the outer leaflet of the flipped v-SNARE cells. This manipulation did not affect the fusion rate or pore properties (data not shown), consistent with the lack of effects in Giraudo et al., 2006) in the absence of synaptotagmin. Thus, SNARE-only fusion can tolerate a range of target membrane compositions.

*3) Another limitation of this study is that it is focuses on SNAREs only. Prior experiments have been done with a combination of large and small fluorescent probes to measure fusion pore dilation, e.g.: (Lai et al., 2013.), and concluded that factors such as synaptotagmin are required for efficient pore expansion. Moreover, with the help of these other proteins, the number of SNARE complexes needed for fusion could be much smaller. The authors do mention accessory proteins at the end of the manuscript, but only in the context of organizing SNARE complexes, as indicated by the term 'accessory' itself, and without considering a direct role in membrane fusion. At the minimum, the authors should tone down every conclusion regarding the number of SNAREs required for neurotransmitter release.*

The question of copy number requirements for SNAREs during fusion has been an ongoing debate for some time, with essentially all in vitro studies having been based on experiments with SNAREs alone. in vivo studies suggested 2-16 SNARE complexes may be required for exocytosis, whereas SNARE-alone in vitro experiments suggested 1 to more than 20 SNAREs may be needed. The fact that similar ranges have been found for in vivo and in vitro studies suggests that the SNARE-alone in vitro studies may indeed have some relevance to exocytosis, but the broad range of values reported precludes any firm conclusions. Only one study offered a possible explanation for the broad range of SNARE requirements; smaller liposomes required fewer SNAREs to fuse than larger ones, suggesting curvature may be an important parameter (Hernandez…Jahn, PNAS 2014, 111:12037-12042). In this respect, our studies of SNARE-alone fusion are highly relevant, as they suggest i) pore expansion requires many more SNAREs than does pore opening, a point that was not suspected previously, ii) when copy number requirements are studied using small soluble probes (or lipid labels) that can permeate small pores, lower number requirements are are likely to be found, iii) curvature effects alone cannot explain copy number requirements (as both the disc and plasma membranes are rather flat in our studies).

The term “accessory proteins” is not intended in any way to diminsh the importance of the other proteins involved in exocytosis. Some of these, such as Munc18, are as essential as SNAREs for exocytosis. However, the SNAREs hold the core position in the fusion machinery because of their ability to induce fusion (albeit inefficient and slow) when reconstituted alone. The other constituents of the fusion machinery, regardless of whether they are required in vivo, cannot hold the same claim.

We changed the word “accessory” to “other”, and have modified the Introduction (see below) and Discussion extensively to clarify that mutations in Munc18, Synaptotagmin, and Complexin affect fusion pore properties. Therefore, all these, and likely other components of the exocytotic machinery are important regulators of fusion pores. Because we have not yet tested the roles of these components, one should be cautious about extrapolating our results to physiological neurotransmitter or hormone release. Nevertheless, the concept of protein crowding driving pore dilation is such a general principle that is likely to apply to the physiological situation qualitatively. Indeed, there are some suggestions in the literature that SNARE availability affects release kinetics (Acuna…Sudhof, Neuron, 2014, and Zhao[…]Lindau, PNAS 2013).

We now cite Lai et al. along with other relevant work in the Discussion. Lai et al. found pore dilation to be surprisingly slow in SNARE-only fusion in a tethered single-liposome fusion assay. Inclusion of Synaptotagmin, Ca^2+^ and complexin increased the speed (to ~10 s) and efficiency of content mixing significantly (to ~5% of docked liposomes). Increasing SNARE copy numbers increased content mixing with the small probe sulforhodamine B, but the effect on the permeability to the large DNA cargo was not investigated. Our results are consistent with the Lai et al. study. We also expect Synaptotagmin and other components of the fusion machinery to modify pore properties in our assay, as their mutations do during exocytosis. Just like the reviewers, we expect Synaptotagmin may reduce the SNARE copy requirement for pore dilation to smaller numbers. We also expect the more copies of Syt-SNARE complexes are involved, the easier will pore dilation be. Incorporating these other important exocytotic proteins into the assay will be an exciting and important future direction.

*4) Please indicate the percentage of vNLPs that fuse with the cells for the different vNLPs. This is a critical issue when rationalizing the data in terms of how the number of SNAREs influence fusion pore properties because the vNPLs are expected to have a distribution of VAMP copies rather than a single number. If the percentage of vNLPs that is low, the fusion may arise from the population that has much higher VAMP copies than assumed from the average. Describing the percentage of vNPLs that fuse is also important to evaluate this overall approach.*

In response to this excellent point made by the reviewer, we performed a new set of experiments, which showed that the fraction of docked NLPs that fuse is of order 4-5% over 20 min. These new results are presented in Figure 2—figure supplement 1. These new experiments are similar to our single-cell fluorescence measurements that probed lipid mixing (Figure 2), in which we include a pair of fluorophores, DiI and DiD, in the NLPs. We incubated these fluorophores with flipped t-SNARE cells and monitored DiI (donor) fluorescence over time, as before. In the new experiments, however, at the end of the experiment, we bleached the acceptor DiD completely to obtain the maximum possible donor (DiI) intensity, Fmax We then rescaled the donor fluorescence values F˜(t)=(F−Fo)/(Fmax−Fo), where Fo is the minimum at the beginning of acquision. This analysis revealed that 4-5% of maximal lipid mixing was reached at the end of 20 minutes of acquisition. (This is likely an underestimate of the true fraction that fuses, as some fusion inevitably occurs during the few minutes between the time the 4°C fusion block is removed and acquisition starts on the 37°C stage-top incubator of the microscope). By comparison, fusion between v-SNARE NLPs and t-SNARE liposomes yields a similar extent of lipid mixing over ~20 min (Bello et al., 2016).

The fact that the fusion rate does not increase for mean copy number greater than 2-4 per NLP side (Figure 5) strongly suggests that there is no bias for higher copy numbers (assuming docking is comparable for all samples). Were there such a bias, i.e. higher copy numbers fused more readily, we would then expect the fusion rate and hence the fused faction to increase with mean copy number, since higher mean copy number means that more NLPs are present with large copy numbers.

Even if there were such a bias for high copy numbers, our results would still correctly report the general trend of pore properties versus copy number. This can be seen from the following argument. A reasonable guess for the distribution of VAMP copy numbers per NLP, n, is a Poisson distribution, p(n)=e−n¯(n¯)n/n!, provided the mean copy number n¯ is well below 30, the maximum attainable value in our experiments (presumably a packing constraint). In our experiments, we fix n¯ to have various values. For example, consider experiments with a mean of 4 copies per NLP, n¯=4; then even if the 5% of NLPs that fuse correspond to only the largest copy numbers, this would mean that most fused NLPs would have n = 8, 9 or 10 SNAREs (because ∑n=830p(n) ≈ 5%, and the biggest contributions are from 8 ≤ n ≤ 10). Similarly, for n¯ = 8 most fused NLPs would have 13 ≤ n ≤ 15. Thus, even if there is the bias that the reviewer fears, the typical copy number of the NLPs whose pore properties are measured is an increasing function of the mean value n¯. It’s just that n¯ underestimates somewhat the actual copy numbers involved.

Let us also consider the extreme limit where fusion might arise from NLPs bearing the maximum possible copy number of v-SNAREs, ~30, due to packing constraints. If this were the case, then as the mean copy number increased, the fusion rate should also increase (inconsistent with observations of Figure 5), but every pore would have similar properties (since every pore would be induced by ~30 SNARE copies). This is also inconsistent with data (Figure 5 and its supplement 1).

*5) The number of fusion pores/min observed with empty NLPs (is what eNLP means?) is low compared to that observed with vNLP8, but not negligible; it appears to be about 8 times smaller in Figure 4. Does this mean that eNLPs fuse spontaneously with the cells? If this is the case, an acceleration by a factor of 8 suggests that 8 SNARE complexes only provide about 1 kcal/mol to facilitate membrane fusion. If there is no flaw in this argument, the authors should emphasize this point and compare this energetic estimate will all the energetic arguments they make later in the manuscript.*

Yes, eNLP means empty NLP, as defined in Results, paragraph two.

The level of activity that we see with eNLPs is about the same as the “baseline” or “background” level of activity that we measure when NLPs are omitted altogether. (This is true of all the negative controls we test, eNLP, TeNT, etc., see Wu et al., 2016). Now the pores in the absence of NLPs cannot be fusion pores, which form from two apposed bilayers, but are instead likely to be simple bilayer pores in the plasma membrane, leaks around the patch, or any other artifact that results in a small transient current. Thus, it is likely that eNLPs do not produce fusion pores.

For this reason, we believe that the reviewer’s argument leading to ~ 1 kcal/mol is actually not applicable: whatever their origin, because these background currents are not fusion pores, SNARE-reconstituted NLPs do not just accelerate the rate of occurrence of fusion pores. SNAREs induce fusion, a qualitatively different event.

Regarding the energetic arguments made later in the manuscript, we wish to emphasize that these refer to an already nucleated fusion pore, and describe the relative energies of pores of different sizes. These relative energies are quite different to the energy governing the rate of fusion, namely the energy to nucleate a fusion pore in the first place.

*6) Conclusions regarding fusion pore expansion should be taken with caution because the scaffolding protein may impose constraints that would not be present* in vivo.

We agree, and we now allude to a potential effect of this constraint on pore lifetimes in the modified Discussion. However, we note that we chose to use large NLPs rather than smaller, MSP-based discs so that the scaffold would not hinder pore dilation up to large pore sizes (at least up to ~5 nm radius). Indeed, most pores do not reach large enough sizes to experience constraints imposed by the scaffold proteins.

*7) A concern is about the possibility of formation of multiple pores. In the third paragraph of the Results, the authors provided an explanation for single pore formation. However, they apparently excluded the possibility of pore formation involving multiple v-SNARE nanolipoprotein particles at the same time. Due to the large size of nanolipoprotein particles (23 nm in diameter), it is possible that there is more than one fusion pore for individual nanolipoprotein particle in the presence of many SNAREs. For example, compared to the large jump from 15 to 30 v-SNAREs, the difference between 8 and 15 v-SNAREs is negligible (Figure 5). Could this imply the formation of another fusion pore?*

We believe that the occurrence of multiple pores per NLP is unlikely for the following reasons.

i) In previous work we studied pores in MSP based nanodiscs, which can accommodate only a single pore due to their limited size. We showed that pores connecting MSP discs (~15 nm diameter, bearing 8-9 total copies of v-SNAREs) and flipped t-SNARE cells are not permeable to a large ion, NMDG+ (Wu et al., 2016), likely because the maximal size of such pores is constrained by the scaffold protein (see Figure 7, energy profile labeled “vMSP”).

If a larger NLP bearing the maximally allowed SNARE copies had multiple small pores each similar to the pore seen in MSP discs, these pores would similarly be impermeable to NMDG+. If instead there were a single large pore per NLP, then NMDG+ could permeate the pore. To address this question, we performed new experiments in which we measured the permeability of vNLP30 pores to NMDG+. In contrast to MSP nanodiscs the large ion was permeant through these vNLP30 pores. These new results are presented in Figure 5—figure supplement 4.

These results are consistent with those of Bello et al., 2016 who showed that progressively larger cargo could be released from t-SNARE liposomes during fusion with vNLPs as the v-SNARE copies per NLP was increased.

ii) Conductance of n small pores in a single NLP would be roughly additive, giving total conductance equal to gpo=n x gpo, where gpo is the mean open-pore conductance of a small pore. Doubling the SNARE copies would presumably at most double n, and by consequence, total conductance. The fact that we find faster than linear increase in mean pore conductance as a function of copy number (Figure 5) is consistent with each NLP bearing a single pore whose size increases with increasing SNARE copies (recall that conductance is proportional to the cross- sectional area of the pore, ∝ rpore2 i.e. very sensitive to changes in pore size).

iii) If multiple small pores occurred per NLP, this should be evident in the distribution of point- by-point conductance values, with peaks at n × gpo, where n = 1, 2, 3 …. Instead, for the distribution of mean gpo for vNLP30 we find a peak at ~300 pS, and a broad peak at ~3-14 nS (Figure 6). If the typical small pore has 300 pS conductance, then to have ~6 nS (typical large conductance), there would have to be ~20 small pores per NLP. It is hard to imagine this many pores coexisting in this small area.

iv) If the observed conductance increase with increasing SNARE copy number were due to an increasing number of small pores in each NLP, we might expect the fusion rate we measure (the number of bursts per unit time) to increase with copy number. On the contrary, we find the rate saturates above ~ 4 copies. While we cannot exclude that the multiple pores could be occurring simultaneously, this saturation strongly suggests the number of pores is not increasing.

In conclusion, although we cannot rule out that occasionally a small number of pores simultaneously appear in a single NLP, all the evidence suggests this is an infrequent event.

We added these arguments in the manuscript.

*8) In the section 'Derivation of best-fit model parameters[…]', the authors showed how they obtain the parameters in the model by fitting the calculated results of the model (equation 11) to the experimental data. For example, they fit the results in the range of 0.2 nm ≤*
rpo
*≤ 1.5 nm to obtain the tension of membrane γ and the steric-hydration force length scale λ (0.5 nm ≤*
rpo
*≤ 2.5nm) to determine a range for epsilon (4 nm ≤ rpo ≤ 4.5nm) to obtain tau. However, in equation 11, the total free energy depends on all the parameters in the full range of*
rpo*, so how can the authors obtain an individual parameter in a different range of*
rpo*?*

The different parameters in the model exert their strongest influence over different ranges of pore sizes and SNARE copy number. For each parameter, we fix its value to produce a best fit of model predictions to experiment in its dominant range. Once parameters are set in this way, we compare model predictions and experimental data for all values of pore size, with good agreement over the entire range (Figure 7). This scheme tests our model more stringently, we believe, than simultaneously choosing all parameters to globally optimize the fit to all data.

For example for the membrane parameters (tension γ, hydration length λ and bending modulus κ) it is best to fit to pure membrane data. The closest we can get to this is 1 SNARE data (vNLP1, Figure 7). Thus, we choose λ to reproduce the location of the minimum of the experimental vNLP1 curve, as these smallest scales are dominated by hydration effects. As the tension γ largely determines the slope beyond this minimum, we chose γ to fix this slope by fitting the model in the region 0.5 nm ≤ rpo ≤ 1 nm (i.e., from the minimum to the limit of the vNLP1 data). Finally, the bending modulus κ is a fixed material parameter whose value we use from previous experimental measurements in the literature.

As SNAREs are introduced, the model predicts the minimum barely shifts, while the slope beyond the minimum decreases with increasing numbers of SNAREs by an amount depending on the zippering energy parameter ϵ_zip_. Thus we selected a typical experimental curve (vNLP30) and chose ϵ_zip_ to reproduce this slope beyond the minimum up to the limit of the data (0.5 nm ≤ rpo≤ 2.5 nm).

Finally, for pore sizes beyond ~2.5 nm the ApoE proteins are twisted and resist further expansion in our model (Figure 7 and Figure 7—figure supplement 1). Only in this region is the free energy profile affected by τ, the torque per unit length exerted by these proteins. Thus, we obtained τ as a best-fit parameter by comparing slopes for large pore sizes (> 4 nm) where twisting is the dominant effect resisting expansion.

To help clarify this overall approach, we modified “Derivation of best-fit model parameters by fitting model-predicted free energy to experiment” in Materials and methods to include a better description of this procedure.

Please note. In the originally submitted manuscript we wrongly used the vMSP curve of Figure 7 to fix the membrane parameters γ, λ. In the revised manuscript we used the correct curve, vNLP1 (the closest to membrane-free). Thus, the best fit parameter values were updated; they differ typically by only ~20% from those presented in the original manuscript.

*9) The parameters in the mathematical model (i.e., rpo, h, δ, D, and the twisting angle phi of ApoE proteins) should be explicitly shown in the model in Figure 7.*

Thank you for this excellent suggestion. We updated the schematic of Figure 7 (and caption) so that these quantities are now explicitly indicated (this is our understanding of the reviewer’s suggestion). To further convey the nature of *ϕ*, the twist angle of the ApoE proteins, we also updated Figure 7—figure supplement 1 to include the dependence of *ϕ* on pore size rpo.

*10) To calculate the second term in equation 7, the authors approximated the pore as a cylinder of radius rpo and height λ = 0.13 nm. It seems that the height of the cylinder is too small, as the neck of the fusion pore can be on the order of ~5 nm in height. Please explain why choosing such a small height for the cylinder is a reasonable approximation.*

We updated “Free energy contribution from short-ranged steric-hydration forces” of Materials and methods to explain this point, as follows.

The steric hydration interaction has a very short spatial range, of order the hydration scale λ. The biggest contribution to the energy comes from the narrowest part of the pore, the waist of diameter 2rpo. In addition, a height *l* of pore such that the pore diameter increases to 2rpo + λ will contribute; parts of the pore further away than *l* contribute negligibly because the pore diameter is much greater than 2rpo + λ at those locations, so the hydration energy will have decayed to negligible values relative to its value at the waist. The value of *l* is very small (much smaller than the true pore height *h*) because λ is very small.

Please note: on re-examining this point in response to the reviewer’s query, we discovered that we had erroneously concluded that the height of the pore that contributes is equal to the hydration length, *l* = λ. In fact, the correct expression is *l* = l=2λ(H+2δ where δ is the thickness of the membrane. We updated “Free energy contribution from short-ranged steric- hydration forces” to reflect this correction and redid our procedure to obtain λ as the best-fit parameter. The best-fit value is now λ = 0.10 nm (cf. 0.13 nm previously), Figure 7—figure supplement 2. The figures presenting our model results were accordingly updated; most are almost identical to the original figures (Figure 7, Figure 7—figure supplement 1).

[Editors' note: further revisions were requested prior to acceptance, as described below.]

*The manuscript has been improved but there are two remaining issues that need to be addressed before acceptance, as outlined below, using the same bullet numbers used for the points of the decision letter.*

*2) Some of the arguments offered with regard to lipid composition are reasonable, but saying that 'short PI(4,5)P2 does not have an effect (data not shown)' is not convincing, and the problem still remains that it is difficult to control lipid composition in this system and make it similar to physiological. This is particularly important considering the low percentage of fusion that they now report (point 4). The authors cite Giraudo et al., 2006, but the work in that paper suffers from the same problem. This concern does not invalidate the results from the authors, but they should acknowledge the problem in the manuscript. The fact is that lipids could play key roles in the fusion mechanism and the tendency of SNARE-centric models of membrane fusion to ignore this fact is deleterious to the scientific discussion in the field.*

We agree with the reviewer that our assay does have the limitation that the lipid compositions involved do not match physiological lipid compositions. Accordingly, we reworded the paragraph that discusses this issue (subsection “A few SNARE complexes are sufficient to create a fusion pore, but many more are needed to dilate it”). This paragraph now explicitly states this limitation:

“In contrast, in our vNLP-tCell fusion assay, the target membrane is the outer leaflet of the plasma membrane which is largely devoid of negatively charged lipids. In general, a limitation of our system is that the lipid composition of the outer leaflet of the t-SNARE-presenting cell differs substantially from that of the plasma membrane inner leaflet, and lipid composition can play a key role in fusion.”

In addition, the final sentence of the paragraph regarding the tolerance of SNARE- only fusion to various lipid compositions was softened. It now reads:

“This swap resulted in similar fusion rates and pore properties for two different SNARE copy numbers per NLP (Figure 5—figure supplement 3), suggesting that fusion mediated by SNAREs alone may not be very sensitive to target membrane composition within a certain range.”

*4) The authors again make some good arguments, but they really do not address the heart of the problem. With such low percentage of fusion, it seems very likely that fusion occurs for low populations of nanodiscs with higher copy number of SNAREs than the mean. The authors claim that this is unlikely because 'the fusion rate does not increase for mean copy number greater than 2-4 per NLP side (Figure 5)'. However, in Figure 5 one can see a tendency for the fusion rate to increase up to vNLP15. Even though the differences may not be statistically significant, one just cannot draw the conclusion written by the authors. Hence, the authors should explicitly acknowledge that the actual numbers of SNAREs underlying fusion may be (in fact that are very likely to be) higher than the mean values described. The same issue applies to measurements of conductance.*

In response to the reviewer’s comment, we statistically compared the fusion rates for copy numbers equal to and greater than 4. (In the previous version of the manuscript we compared these fusion rates to the value for zero copy number, i.e. SNARE-free discs, but we did not compare them to one another). This analysis shows definitively that there are no statistically significant differences between the fusion rates for vNLP4, vNLP8, vNLP15 and vNLP30. Thus, there is no evidence for a bias to copy numbers higher than the mean value arising as a result of higher fusion rates for higher copy numbers.

We made the following changes to the manuscript. (1) This statistical analysis is now depicted in Figure 5, mentioned in its caption, and the source data and statistical analyses are uploaded as a zipped source data file for the figure. (Chernomordik and Kozlov, 2008) We added a new paragraph (Discussion, paragraph three) mentioning and interpreting this statistical result. (3) The new paragraph also presents a condensed version of the simple argument we presented in our previous rebuttal, based on an assumed Poisson distribution of copy numbers. This argument shows that, even if such a bias were present, the copy numbers that fuse would still increase with the mean copy number so that the general trends we report would remain valid. (4) This new paragraph also includes an acknowledgement that the bias feared by the reviewer remains a possibility: “While we cannot categorically exclude this possibility, we think it unlikely.”

While carrying out the new analysis, we discovered an error in the calculation of the fusion rate for the vNLP1 samples (Figure 5). The error resulted in an estimate twice the actual rate for this category only. This error did not change any of our conclusions (the fusion rates are not significantly different among the vNLP4, vNLP8, vNLP15 and vNLP30 samples, at least two SNAREs per face are needed for significant fusion, etc.). We updated the figure and our analysis using the correct rate. We also uploaded the analysis files (which include raw data) as source data, as mentioned above.